# H3K4 mono- and di-methyltransferase MLL4 is required for enhancer activation during cell differentiation

**Ji-Eun Lee[1†], Chaochen Wang[1†], Shiliyang Xu[1,2], Young-Wook Cho[1,3], Lifeng Wang[1], Xuesong Feng[4], Anne Baldridge[1], Vittorio Sartorelli[4], Lenan Zhuang[1], Weiqun Peng[2,5]*, Kai Ge[1]***

[1]Adipocyte Biology and Gene Regulation Section, Laboratory of Endocrinology and Receptor Biology, National Institute of Diabetes and Digestive and Kidney Diseases, National Institutes of Health, Bethesda, United States; [2]Department of Physics, The George Washington University, Washington, United States; [3]Chuncheon Center, Korea Basic Science Institute, Chuncheon, Republic of Korea; [4]Laboratory of Muscle Stem Cells and Gene Regulation, National Institute of Arthritis, Musculoskeletal, and Skin Diseases, National Institutes of Health, Bethesda, United States; [5]Department of Anatomy and Regenerative Biology, The George Washington University, Washington, United States

**\*For correspondence:**
kaig@niddk.nih.gov (KG);
wpeng@gwu.edu (WP)

[†]These authors contributed equally to this work

**Competing interests:** The authors declare that no competing interests exist.

**Reviewing editor**: Christopher Glass, University of California, San Diego, United States

**Abstract** Enhancers play a central role in cell-type-specific gene expression and are marked by H3K4me1/2. Active enhancers are further marked by H3K27ac. However, the methyltransferases responsible for H3K4me1/2 on enhancers remain elusive. Furthermore, how these enzymes function on enhancers to regulate cell-type-specific gene expression is unclear. In this study, we identify MLL4 (KMT2D) as a major mammalian H3K4 mono- and di-methyltransferase with partial functional redundancy with MLL3 (KMT2C). Using adipogenesis and myogenesis as model systems, we show that MLL4 exhibits cell-type- and differentiation-stage-specific genomic binding and is predominantly localized on enhancers. MLL4 co-localizes with lineage-determining transcription factors (TFs) on active enhancers during differentiation. Deletion of *Mll4* markedly decreases H3K4me1/2, H3K27ac, Mediator and Polymerase II levels on enhancers and leads to severe defects in cell-type-specific gene expression and cell differentiation. Together, these findings identify MLL4 as a major mammalian H3K4 mono- and di-methyltransferase essential for enhancer activation during cell differentiation.

## Introduction

Enhancers are gene regulatory elements critical for cell-type-specific gene expression in eukaryotes (*Bulger and Groudine, 2011*). Lineage-determining or signal-dependent transcription factors (TFs) bind to enhancers and recruit chromatin modifying and remodeling enzymes and the Mediator coactivator complex to initiate RNA polymerase II (Pol II)-mediated transcription at promoters (*Roeder, 2005*). Recent genome-wide analyses of histone modifications, binding of transcription coactivators p300 and MED1 and lineage-determining TFs, coupled with functional assays of gene regulatory elements, have enabled the identification of enhancer chromatin signatures (*Heintzman et al., 2007*; *Lupien et al., 2008*; *Wang et al., 2008*; *Heintzman et al., 2009*; *Visel et al., 2009*; *Creyghton et al., 2010*; *Ghisletti et al., 2010*; *Heinz et al., 2010*; *Ernst et al., 2011*; *Rada-Iglesias et al., 2011*). In contrast to active promoters that are marked by H3K4me3, enhancers are marked by H3K4me1 and H3K4me2 (H3K4me1/2) but little H3K4me3, and are often bound by cell-type-specific TFs. Active enhancers are further marked by

**eLife digest** Almost every cell in a human body carries the same genes, but not every cell will express all of these genes as proteins. As different types of cells, such as brain, liver, fat or muscle cells, develop, they will express different genes; or they will express the same genes, but at different times and in different amounts. Enhancers are short stretches of DNA that boost the amount of protein that is produced when a gene is expressed, and they are particularly important for those genes that are expressed differently between cell types.

Enhancers bolster expression of a gene by interacting with the DNA nearby. Even genes separated from enhancers by a long stretches of DNA can benefit because the way that DNA is tightly packed inside the nucleus means that two distant sequences can actually end up close together. Proteins called transcription factors will bind to enhancers and recruit the cell's protein 'machinery' required to express nearby genes. Enhancers can be identified by specific chemical marks associated with their DNA, but little is known about the enzymes that leave these marks in mammals. Moreover, it is not clear which genes are influenced by these marks.

Now, by examining fat cells and muscle cells as they mature, Lee et al. have found that an enzyme called MLL4 is responsible for adding chemical marks to enhancers in both humans and mice. Further, MLL4 is required both to allow cells to specialize into different cell types, and to boost the expression of genes that are specific to each type of mature cells. Since faulty MLL4 has been implicated in several cancers and developmental defects, the findings of Lee et al. may lead to a better understanding of these diseases.

histone acetyltransferases CBP/p300-mediated H3K27ac (*Creyghton et al., 2010*; *Rada-Iglesias et al., 2011*). H3K4me1 on enhancers often precedes H3K27ac and activation of enhancers. A large number of recent publications have validated the predictive power of such enhancer chromatin signatures in identification of novel enhancers critical for cell-type-specific gene expression and cell differentiation (reviewed in *Calo and Wysocka, 2013*).

In yeast, the Set1 complex is the sole H3K4 methyltransferase. Through the enzymatic subunit SET1, it is responsible for all mono-, di- and tri-methylations on H3K4 (*Ruthenburg et al., 2007*). In Drosophila, there are three Set1-like H3K4 methyltransferase complexes that use dSet1, Trithorax (Trx) and Trithorax-related (Trr) as the respective enzymatic subunits (*Mohan et al., 2011*). Mammals possess six Set1-like H3K4 methyltransferase complexes (*Figure 1—figure supplement 1A*). Based on the sequence homology of enzymatic subunits and the subunit compositions (*Cho et al., 2007*; *Ruthenburg et al., 2007*; *Vermeulen and Timmers, 2010*; *Cho et al., 2012*), they fall into three sub-groups: MLL1/MLL2 (also known as KMT2A and KMT2B, respectively), MLL3/MLL4 (MLL3 is also known as KMT2C; MLL4 is also known as KMT2D, ALR and sometimes MLL2), and SET1A/SET1B (also known as KMT2E and KMT2F, respectively), which correspond to *Drosophila* Trx, Trr and dSet1 complexes, respectively. dSet1 is responsible for the bulk of H3K4me3 in Drosophila (*Ardehali et al., 2011*; *Mohan et al., 2011*). Consistently, depletion of the unique CFP1 subunit of SET1A/SET1B complexes in mammalian cells markedly decreases global H3K4me3 level, suggesting that SET1A/SET1B are the major H3K4 tri-methyltransferases in mammals (*Clouaire et al., 2012*). In contrast, knockdown of Trr, the *Drosophila* homolog of MLL3/MLL4, decreases global H3K4me1 levels, indicating that Trr regulates H3K4me1 in Drosophila (*Ardehali et al., 2011*; *Mohan et al., 2011*). However, the histone methyltransferases (HMTs) responsible for H3K4me1/2 on mammalian enhancers remain elusive. Further, the functions of these H3K4 mono-/di-methyltransferases on enhancers and in regulating cell-type-specific gene induction and cell differentiation are unclear. Finally, how these HMTs are recruited to enhancers needs to be clarified (*Calo and Wysocka, 2013*).

Adipogenesis and myogenesis are robust and synchronized models of cell differentiation. Differentiation of preadipocytes towards adipocytes, that is adipogenesis, is regulated by a network of sequentially expressed adipogenic TFs (*Rosen and MacDougald, 2006*). Peroxisome Proliferator-Activated Receptor-γ (PPARγ) is considered the master regulator of adipogenesis and controls adipocyte gene expression cooperatively with CCAAT/enhancer-binding protein-α (C/EBPα) (*Rosen et al., 2002*; *Lefterova et al., 2008*). The early adipogenic TF C/EBPβ marks a large number of TF 'hotspots' before induction of adipogenesis. C/EBPβ not only controls the induction of PPARγ and C/EBPα expression

but also acts as a pioneer factor to facilitate the genomic binding of PPARγ, C/EBPα and other adipogenic TFs during adipogenesis (*Siersbaek et al., 2011*). Adipogenesis in cell culture is synchronized, with the vast majority of cells in the confluent population differentiating into adipocytes within 6–8 days, thus providing a robust model system for studying transcriptional and epigenetic regulation of gene expression during cell differentiation (*Ge, 2012*). Myogenesis is another robust model system for cell differentiation. Ectopic expression of the myogenic TF MyoD in fibroblasts and preadipocytes is sufficient to induce muscle differentiation program characterized by expression of myogenesis markers such as Myogenin (Myog) and Myosin (*Tapscott et al., 1988*; *Lassar et al., 1991*).

Using adipogenesis and myogenesis as model systems, here we show MLL4 is partially redundant with MLL3 and is required for cell differentiation and cell-type-specific gene expression. By ChIP-Seq analyses, we observe cell-type- and differentiation-stage-specific genomic binding of MLL4. MLL4 is mainly localized on enhancers and co-localizes with lineage-determining TFs on active enhancers during differentiation. We demonstrate that MLL4 is partially redundant with MLL3 and is a major H3K4 mono- and di-methyltransferase in mouse and human cells. Furthermore, MLL4 is required for H3K4me1/2, H3K27ac, Mediator and Pol II levels on active enhancers, indicating that MLL4 is required for enhancer activation. Finally, we provide evidence to suggest that lineage-determining TFs recruit and require MLL4 to establish cell-type-specific enhancers.

## Results

### MLL4 is essential for adipogenesis and myogenesis

Among the six SET1-like H3K4 methyltransferases found in mammals, we initially knocked out *Mll3* and *Mll4* individually in mice using gene trap approaches (*Figure 1—figure supplement 1A–E* and data not shown). *Mll3* knockout (KO) mice died around birth with no obvious morphological abnormalities in embryonic development. *Mll4* KO mice showed early embryonic lethality around E9.5. We then generated *Mll4* conditional KO mice (*Mll4^{f/f}*) (*Figure 1A–B*), which were viable and allowed us to investigate MLL4 function in a tissue-specific manner. Conditional KO of *Mll4* was also verified in cell culture. Deletion of *Mll4* led to the disruption of MLL4 complex in cells (*Figure 1—figure supplement 2A-B*).

To understand the role of MLL4 in adipogenesis and myogenesis, we generated *Mll4^{f/f}*;*Myf5-Cre* mice by crossing *Mll4^{f/f}* with *Myf5-Cre* mice (*Tallquist et al., 2000*). Myf5-Cre specifically deletes genes flanked by loxP sites in somitic precursor cells that give rise to both brown adipose tissue (BAT) and skeletal muscle in the back (*Seale et al., 2008*). *Mll4^{f/f}*;*Myf5-Cre* pups and E18.5 embryos were obtained at the expected Mendelian ratio but displayed marked reduction in back muscles and died immediately after birth due to breathing malfunction that is controlled by muscle groups in the rib cage (*Figure 1C–D* and data not shown). Sagittal sections along the midline of E18.5 embryos were subjected to immunohistochemistry analysis using antibodies against BAT marker UCP1 and muscle marker Myosin (*Figure 1E*). Compared with littermate controls and *Mll3* KO, *Mll4^{f/f}*;*Myf5-Cre* embryos showed marked decreases of BAT and muscle mass, suggesting that MLL4 is required for adipogenesis and myogenesis.

To understand how MLL4 regulates BAT development, we induced adipogenesis in brown preadipocytes in culture. KO of *Mll4* led to a moderate differentiation defect along with a transient up-regulation of *Mll3* expression in the early phase of adipogenesis whereas KO of *Mll3* had no effect on adipogenesis (*Figure 2—figure supplement 1A-F*), suggesting a more prominent role of MLL4 in development and a partial compensation of MLL4 loss by MLL3. To eliminate the compensatory effect, we isolated primary brown preadipocytes from E18.5 *Mll3^{−/−}Mll4^{f/f}* embryos. After SV40T immortalization (*Wang et al., 2010*), cells were infected with adenoviral Cre (Ad-Cre) or GFP (Ad-GFP) to generate *Mll3^{−/−}Mll4^{−/−}* and *Mll3^{−/−}* cells, respectively. Deletion of *Mll4* by Cre from the immortalized *Mll3^{−/−}Mll4^{f/f}* brown preadipocytes had little effect on cell growth (*Figure 2—figure supplement 1G-H*), but led to severe defects in adipogenesis and associated expression of adipogenesis markers and key regulators *Pparg* and *Cebpa* as well as brown adipocyte markers *Prdm16* and *Ucp1* (*Figure 2A–B* and *Figure 2—figure supplement 1I*), indicating that MLL3 and MLL4 are functionally redundant and are essential for adipogenesis. Furthermore, ectopic expression of PPARγ or C/EBPβ failed to rescue adipogenesis in *Mll3/Mll4* double KO preadipocytes (*Figure 2—figure supplement 1J* and *Figure 2C*). Consistent with the results from brown preadipocytes, knockdown of *Mll4* in 3T3-L1 white preadipocytes inhibited adipogenesis (*Figure 2—figure supplement 1K–M*). Furthermore, MLL3 and MLL4

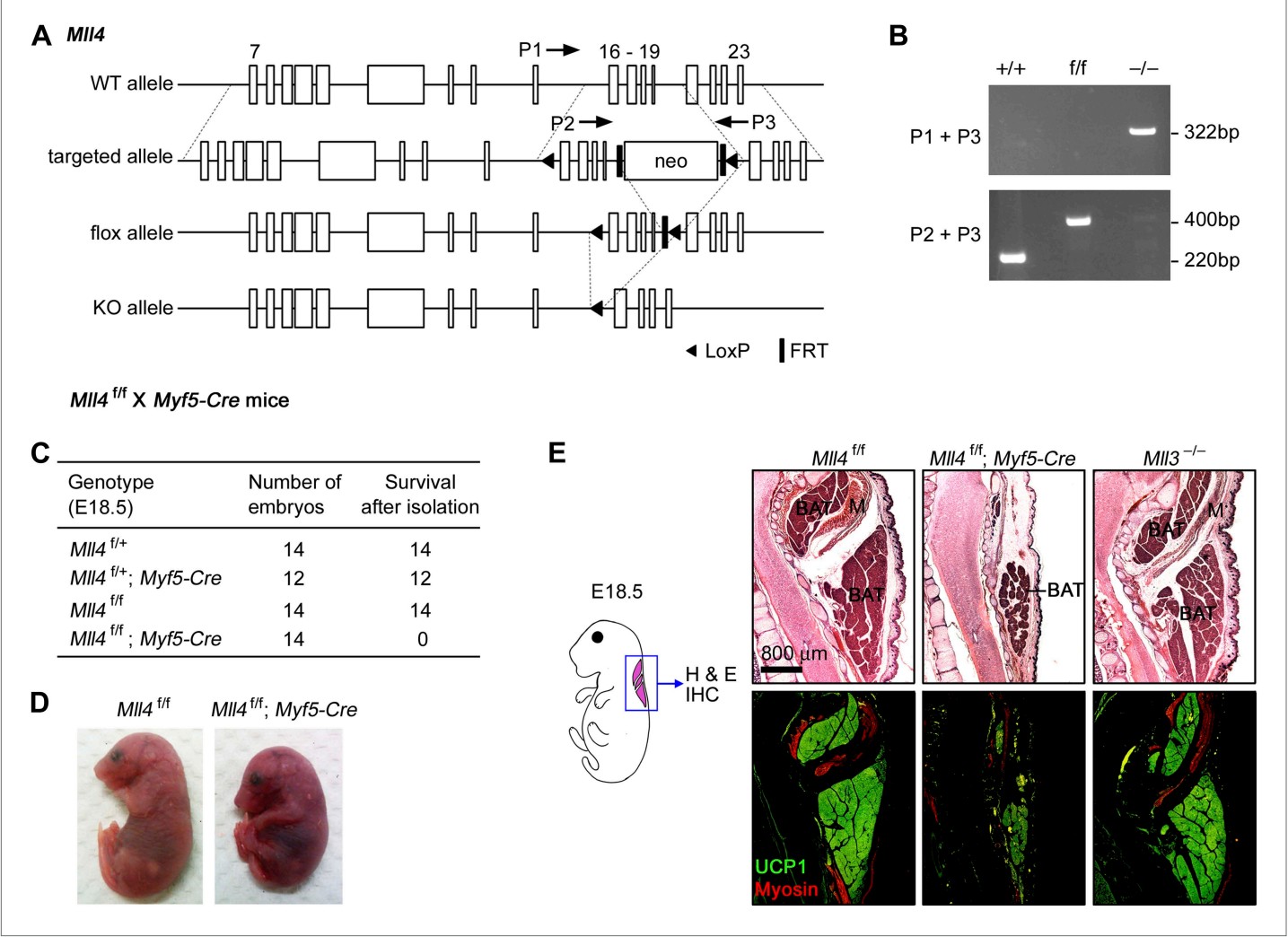

**Figure 1**. MLL4 is required for brown adipose tissue and muscle development. (**A** and **B**) Generation of *Mll4* conditional KO mice (*Mll4^f/f^*). (**A**) Schematic representation of mouse *Mll4* wild-type (WT) allele, targeted allele, conditional KO (flox) allele and KO allele. In the targeted allele, a single loxP site was inserted in the intron before exon 16. A neomycin (neo) selection cassette flanked by FRT sites and the second loxP site was inserted in the intron after exon 19. The locations of PCR genotyping primers P1, P2, and P3 are indicated by arrows. (**B**) PCR genotyping of cell lines using mixtures of P2 + P3 or P1 + P3 primers. The genotypes are indicated at the top. (**C**) Genotype of E18.5 embryos isolated from crossing *Mll4^f/+^;Myf5-Cre* with *Mll4^f/f^* mice. The expected ratios of the four genotypes are 1:1:1:1. *Mll4^f/f^;Myf5-Cre* mice died immediately after cesarean section because of breathing malfunction due to defects in muscles of the rib cage. (**D**) Representative pictures of E18.5 embryos of the indicated genotypes. (**E**) E18.5 embryos were sagittally sectioned along the midline. The sections of the cervical/thoracic area indicated in the schematic were stained with H&E (upper panels) or with antibodies recognizing BAT marker UCP1 (green) and skeletal muscle marker Myosin (red) (lower panels). Scale bar = 800 µm.

The following figure supplements are available for figure 1:

**Figure supplement 1**. Generation of *Mll3* and *Mll4* whole body KO mice.

**Figure supplement 2**. Confirmation of *Mll4* deletion.

were also required for PPARγ-stimulated adipogenesis in mouse embryonic fibroblasts (MEFs) (*Figure 2—figure supplement 1N–P*).

To understand how MLL4 regulates myogenesis, we used retroviral vector to stably express ectopic MyoD in *Mll3^−/−^Mll4^f/f^* brown preadipocytes. Cells were then infected with adenoviral Cre or GFP, followed by induction of myogenesis (*Figure 2F–H*). We found that deletion of *Mll4* from *Mll3^−/−^Mll4^f/f^* brown preadipocytes led to severe defects in MyoD-stimulated myogenesis of preadipocytes and the

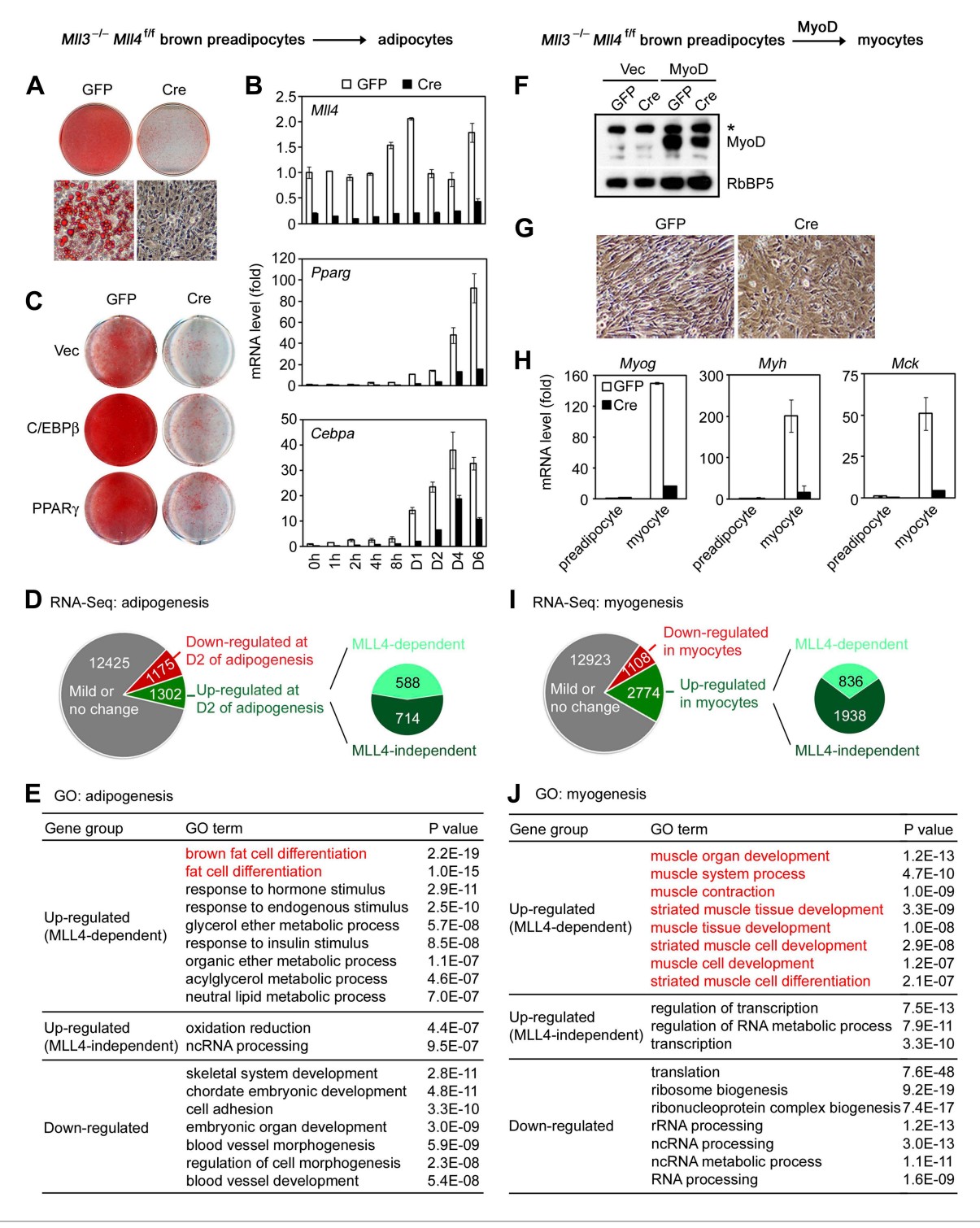

**Figure 2**. MLL4 controls induction of cell-type-specific genes during differentiation. (**A–E**) Adipogenesis of *Mll3⁻/⁻Mll4ᶠ/ᶠ* brown preadipocytes. (**A** and **B**) MLL4 is required for adipogenesis. Immortalized *Mll3⁻/⁻Mll4ᶠ/ᶠ* brown preadipocytes were infected with adenoviral GFP or Cre, followed by adipogenesis assay. (**A**) 6 days after induction of differentiation, cells were stained with Oil Red O. Upper panels, stained dishes; lower panels, representative fields under microscope. (**B**) qRT-PCR of *Mll4*, *Pparg* and *Cebpa* expression at indicated time points of adipogenesis. Quantitative PCR data in all figures are presented as means ± SD. D1, day 1. (**C**) MLL4 is required for C/EBPβ- and PPARγ-stimulated adipogenesis. *Mll3⁻/⁻Mll4ᶠ/ᶠ* brown preadipocytes were infected with retroviruses expressing vector (vec), C/EBPβ or PPARγ. After hygromycin selection, cells were infected with adenoviral GFP or Cre, followed
*Figure 2. Continued on next page*

*Figure 2. Continued*

by adipogenesis assay. (**D**–**E**) MLL4 is required for induction of cell-type-specific genes during adipogenesis. Adipogenesis was done as in (**A**). Cells were collected before (day 0) and during (day 2) adipogenesis for RNA-Seq. (**D**) Schematic of identification of MLL4-dependent and -independent up-regulated genes during adipogenesis. The threshold for up- or down-regulation is 2.5-fold. (**E**) Gene ontology (GO) analysis of gene groups defined in (**D**). (**F**–**J**) MLL4 is required for MyoD-stimulated myogenesis. Immortalized *Mll3−/−Mll4*^f/f brown preadipocytes were infected with retroviruses expressing Vec or MyoD. After hygromycin selection, cells were infected with adenoviral GFP or Cre, followed by myogenesis assay. (**F**) Western blot analysis of MyoD expression before differentiation. RbBP5 was used as a loading control. The asterisk indicates a non-specific band. (**G**) 5 days after induction of differentiation, cell morphologies were observed under microscope. (**H**) qRT-PCR analysis of myogenic gene expression after differentiation. (**I** and **J**) MLL4 is required for induction of cell-type-specific genes during myogenesis. Brown preadipocytes and myocytes were collected for RNA-Seq. (**I**) Schematic of identification of MLL4-dependent and -independent up-regulated genes during myogenesis. The threshold for up- or down-regulation is 2.5-fold. (**J**) GO analysis of gene groups defined in (**I**).

The following figure supplements are available for figure 2:

**Figure supplement 1**. MLL4 is required for adipogenesis.

**Figure supplement 2**. Confirmation of RNA-Seq data by qRT-PCR.

associated expression of myogenesis markers such as *Myog*, *Myh* and *Mck*. Taken together, these data indicate that MLL4 is essential for adipogenesis and myogenesis.

## MLL4 controls induction of cell-type-specific genes during differentiation

Next, we investigated how MLL4 regulates gene expression during differentiation. Adipogenesis was induced in Ad-GFP- or Ad-Cre-infected *Mll3−/−Mll4*^f/f brown preadipocytes. Samples were collected before (day 0) and during (day 2) adipogenesis for RNA-Seq (**Figure 2D** and **Figure 2—figure supplement 2**). In total there were 14,902 genes expressed in Ad-GFP-infected cells at either time point. Among them, 1,175 (7.9%) and 1,302 (8.7%) showed over 2.5-fold down- and up-regulation, respectively, from day 0 to day 2. Among the 1,302 up-regulated genes, a significant number (588, p=1.4E-268, hypergeometric test) was induced in an MLL4-dependent manner. Strikingly, gene ontology (GO) analysis revealed that only the MLL4-dependent up-regulated gene group was associated with fat cell differentiation (**Figure 2E**), suggesting that MLL4 selectively regulates the induction of adipogenesis genes.

RNA-Seq analysis of myogenesis of MyoD-expressing brown preadipocytes revealed a similar trend. 1,108 (6.6%) and 2,774 (16.5%) genes showed over 2.5-fold down- and up-regulation, respectively, from brown preadipocytes to myocytes (**Figure 2I**). Among the 2,774 up-regulated genes, a significant number (836, p<1E-300, hypergeometric test) was induced in an MLL4-dependent manner. Furthermore, among those gene groups, only the MLL4-dependent up-regulated gene group was highly associated with muscle development (**Figure 2J**). Together, these results indicate that MLL4 is required for induction of cell-type-specific genes during differentiation.

## Cell-type- and differentiation-stage-specific genomic binding of MLL4

To find out whether MLL4 directly regulates the induction of cell-type-specific genes during differentiation, we performed ChIP-Seq of MLL4 in adipogenesis and myogenesis using an anti-MLL4 antibody. To exclude false-positive MLL4 targets as a result of off-target antibody binding, ChIP-Seq was done in both Ad-GFP- and Ad-Cre-infected *Mll3−/−Mll4*^f/f cells (i.e., *Mll3* KO and *Mll3/Mll4* double KO cells). High-confidence MLL4 binding signals were obtained by filtering out non-specific signals observed in MLL4-deficient cells (**Figure 3—figure supplement 1A-E**). ChIP-Seq results were independently verified by quantitative ChIP assays at MLL4+ regions on multiple adipogenesis genes (**Figure 3—figure supplement 2**).

ChIP-Seq of MLL4 in adipogenesis was done at three time points, day 0 (preadipocytes), day 2 (during adipogenesis) and day 7 (adipocytes). We identified 6,937, 14,581 and 25,005 high-confidence MLL4 genomic binding regions at day 0, day 2, and day 7, respectively (**Figure 3A**). The average length of MLL4 binding regions was 350–400bp. Interestingly, MLL4 binding regions changed dramatically from day 0 to day 2 but were largely maintained from day 2 to day 7, suggesting differentiation-stage-specific genomic binding of MLL4 during adipogenesis (**Figure 3A**). ChIP-Seq of MLL4 in

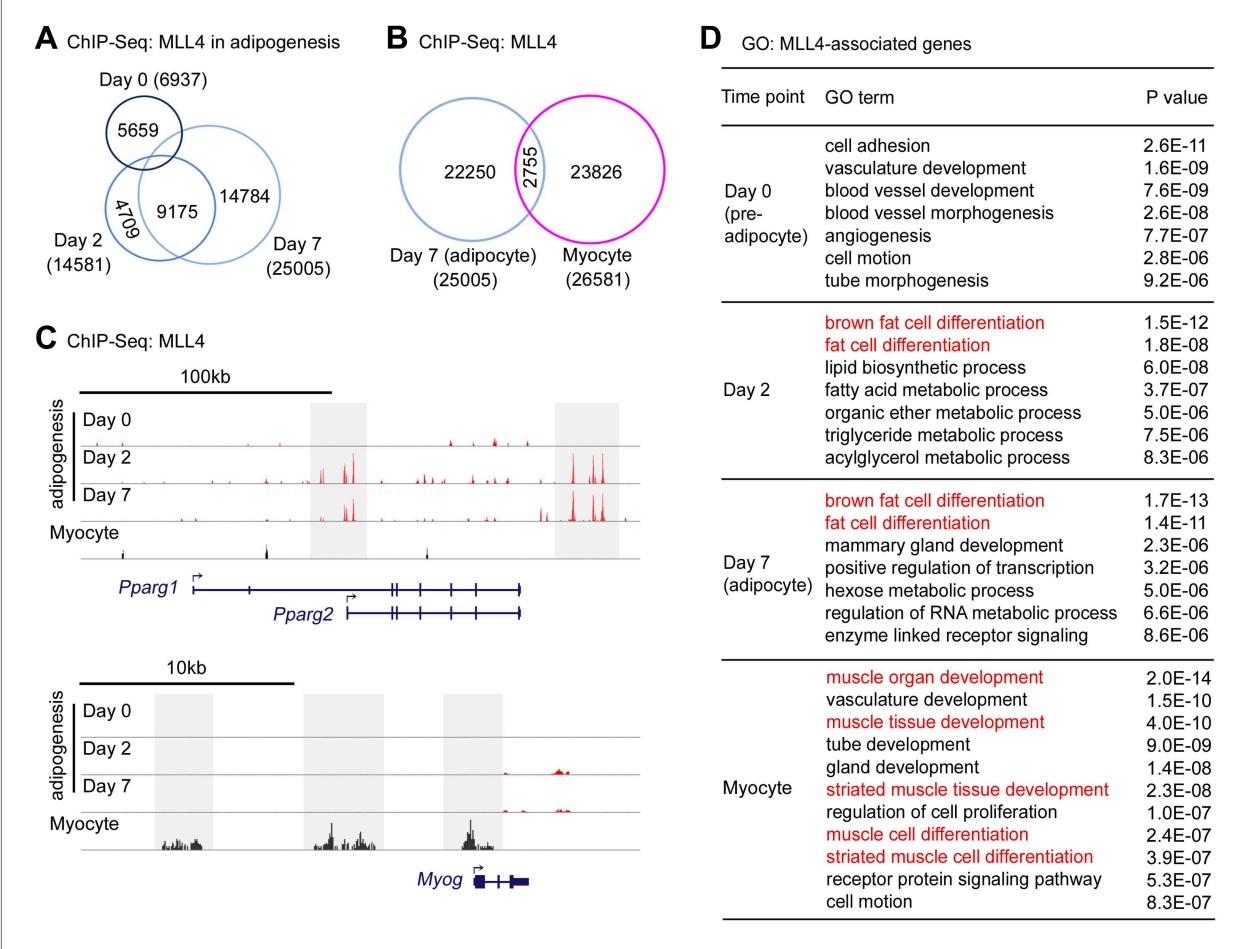

**Figure 3**. Cell-type- and differentiation-stage-specific genomic binding of MLL4. Adipogenesis and myogenesis were done as in *Figure 2A,F,G*, respectively. Cells were collected for ChIP-Seq analysis of MLL4. (**A**) Venn diagram of MLL4 binding regions at day 0 (preadipocytes), day 2 (during adipogenesis) and day 7 (adipocytes) of adipogenesis. (**B**) Venn diagram of MLL4 binding regions in adipocytes and myocytes. (**C**) ChIP-Seq profiles of MLL4 binding on gene loci encoding PPARγ and myogenin (Myog) at indicated time points and cell types. (**D**) GO analysis of genes associated with emergent MLL4 binding regions at indicated time points and cell types.
The following figure supplements are available for figure 3:

**Figure supplement 1**. MLL4 binds to adipogenesis genes.

**Figure supplement 2**. ChIP-qPCR confirmation of ChIP-Seq data.

myocytes identified 26,581 high-confidence MLL4 genomic binding regions. The MLL4-binding regions in adipocytes and myocytes were largely non-overlapping (*Figure 3B*), suggesting cell-type-specific genomic binding of MLL4.

Cell-type- and differentiation-stage-specific genomic binding of MLL4 was confirmed on *Pparg* and *Myog* gene loci that encoded the adipogenic TF PPARγ and the myogenic TF Myog, respectively (*Figure 3C*). MLL4 was also found to bind gene loci encoding other adipogenesis markers such as C/EBPα, KLF15, aP2 (Fabp4), LPL and UCP1 at day 2 and day 7 of adipogenesis (*Figure 3—figure supplement 1F–K*). Interestingly, MLL4 was largely absent from the *Cebpb* gene locus encoding the pioneer adipogenic TF C/EBPβ.

To identify MLL4-associated genes, we used proximity to assign the top 2,000 emergent MLL4 binding regions at each time point to the nearest genes. GO analysis of MLL4-associated genes identified brown fat cell differentiation and fat cell differentiation as the top two functional categories at day 2 (during adipogenesis) and day 7 (adipocytes) but not at day 0 (preadipocytes) or in myocytes

(*Figure 3D*). GO analysis also identified muscle organ/tissue development and muscle cell differentiation as the top functional categories specifically in myocytes (*Figure 3D*). These results are consistent with GO analysis of RNA-Seq data and suggest that cell-type- and differentiation-stage-specific genomic binding of MLL4 is directly involved in the regulation of genes critical for cell differentiation.

## Genomic co-localization of MLL4 with lineage-determining TFs during differentiation

Next, we performed motif analysis of the top 2,000 emergent MLL4 binding regions at each time point (*Figure 4A*). At preadipocyte stage before adipogenesis (day 0), MLL4 binding regions were enriched with motifs of TFs functioning in various developmental lineages. However, at day 2 and day 7 of adipogenesis, MLL4 binding regions were highly enriched with motifs of major adipogenic TFs, including C/EBPα, C/EBPβ, PPARγ, EBF1 and GR (*Rosen and MacDougald, 2006*; *Ge, 2012*). In myocytes, MLL4 binding regions were highly enriched with motifs of myogenic TF MyoD and its binding partner TCF3 (also known as E2A) (*Lassar et al., 1991*), as well as myogenic TFs Runx1 and TEAD4 (*MacQuarrie et al., 2013*).

To experimentally validate the predicted motifs, we performed ChIP-Seq of C/EBPα, C/EBPβ and PPARγ at day 2 of adipogenesis and of MyoD in myocytes. By comparing the genomic localizations of MLL4 with those of C/EBPα, C/EBPβ and PPARγ, we found that consistent with the motif analysis, ~64% of MLL4 binding regions overlapped with those of C/EBPα/β or PPARγ at day 2 of adipogenesis (*Figure 4B–C*). In particular, genomic co-localization of MLL4 with C/EBPβ, C/EBPα, and PPARγ was observed on *Pparg* and *Cebpa* loci at day 2 but not day 0 of adipogenesis (*Figure 4—figure supplement 1 and 2*). In myocytes, 40% of MLL4 binding regions overlapped with those of MyoD (*Figure 4D–E*). Consistent with these results, we observed a physical interaction of MLL4 with C/EBPβ during adipogenesis (*Figure 4F*). We also found that PPARγ associated with MLL3/MLL4-containing H3K4 methyltransferase complex in cells (*Figure 4—figure supplement 3*), which is consistent with the reported direct interaction between PPARγ and MLL3/MLL4 complex (*Lee et al., 2008*). Together, these results indicate significant genomic co-localization of MLL4 with lineage-determining TFs during adipogenesis and myogenesis.

## MLL4 co-localizes with lineage-determining TFs on active enhancers during differentiation

To characterize the genomic features of MLL4 binding regions during differentiation, we performed ChIP-Seq of H3K4me1/2/3, H3K27ac and Pol II during adipogenesis. Then, we used histone marks to define four types of gene regulatory elements: active enhancer, silent enhancer, active promoter, and silent promoter (*Figure 5A*) (*Creyghton et al., 2010*; *Rada-Iglesias et al., 2011*). Average profile plots revealed that at day 2 of adipogenesis, MLL4 and adipogenic TFs C/EBPα, C/EBPβ and PPARγ were enriched on active enhancers (*Figure 5B*). Genomic distribution analyses showed that among the 14,581 MLL4 binding regions at day 2 of adipogenesis, 9,642 (66.1%) and 1,836 (12.6%) were located on active and silent enhancers while only 451 (3.1%) and 29 (0.2%) were located on active and silent promoters, respectively, indicating that MLL4 mainly bound to enhancers and preferentially active enhancers (p<1E-300, binomial test) (*Figure 5C*).

Genomic distribution analyses also revealed preferential localization of C/EBPs and PPARγ on active enhancers during adipogenesis (*Figure 5C*). Interestingly, 57.3% of the C/EBP-binding and 80.0% of the PPARγ-binding active enhancers were MLL4[+], significantly higher than the genome-wide level of co-localization (31.3% for C/EBPs and 56.3% for PPARγ, respectively), indicating that MLL4 preferentially co-localizes with these TFs on active enhancers than on other genomic locations (p<1E-300 for either C/EBPs or PPARγ, hypergeometric test).

At day 2 of adipogenesis, 11,280 active enhancers were bound with adipogenic TFs C/EBPα, C/EBPβ or PPARγ. These 11,280 active enhancers, which we termed adipogenic enhancers, could be clustered into 3 groups: C/EBP[+]PPARγ[−] (C/EBPα- or β-positive but PPARγ-negative, 8,098 active enhancers), C/EBP[−]PPARγ[+] (1,219), and C/EBP[+]PPARγ[+] (1,963) (*Figure 5D*). MLL4 binding was found on 46.9% and 61.7% of C/EBP[+]PPARγ[−] and C/EBP[−]PPARγ[+] adipogenic enhancers, respectively. Remarkably, MLL4 binding was found on 91.3% of C/EBP[+]PPARγ[+] adipogenic enhancers, which were high-confidence adipogenic enhancers (*Lefterova et al., 2008*) (*Figure 5D*).

Similarly, MLL4 and the myogenic TF MyoD were enriched on active enhancers in myocytes (*Figure 5—figure supplement 1A-B*). We identified 8,091 myogenic enhancers that were active

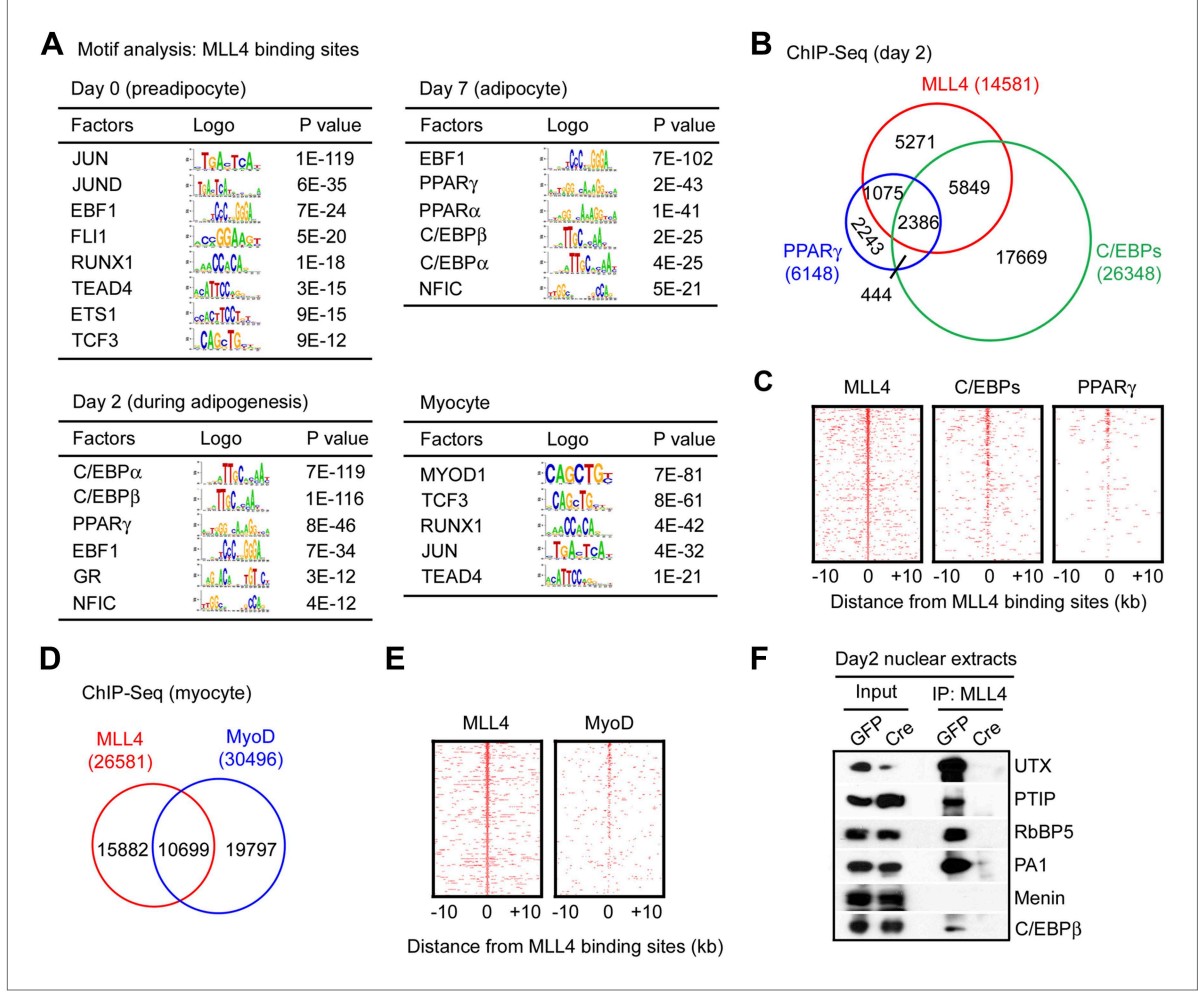

**Figure 4**. Genomic co-localization of MLL4 with lineage-determining TFs during differentiation. (**A**) Adipogenic TF binding motifs are enriched at MLL4 binding regions during adipogenesis while myogenic TF binding motifs are enriched at MLL4 binding regions in myocytes. Top 2,000 emergent MLL4 binding regions at each time point were used for motif analysis. Only TFs that are expressed at the indicated cell type or differentiation stage are included. (**B** and **C**) Venn diagram (**B**) and heat maps (**C**) of genomic co-localization of MLL4 with C/EBPs (C/EBPα or β) and PPARγ at day 2 of adipogenesis. (**D** and **E**) Venn diagram (**D**) and heat maps (**E**) of genomic co-localization of MLL4 with MyoD in myocytes. (**F**) MLL4 physically interacts with C/EBPβ during adipogenesis. Nuclear extracts prepared at day 2 of adipogenesis were immunoprecipitated with MLL4 antibody. The immunoprecipitates were analyzed by Western blot using antibodies against MLL3/MLL4 complex components (UTX, PTIP, RbBP5 and PA1), Menin, or C/EBPβ.

The following figure supplements are available for figure 4:

**Figure supplement 1**. ChIP-Seq and RNA-Seq data on *Pparg* gene during adipogenesis.

**Figure supplement 2**. ChIP-Seq and RNA-Seq data on *Cebpa* gene during adipogenesis.

**Figure supplement 3**. PPARγ interacts with MLL3/MLL4-containing H3K4 methyltransferase complex in cells.

enhancers bound with MyoD. Among the 8,091 myogenic enhancers, 5,228 (64.6%, p<1E-300, hypergeometric test) were MLL4+ (***Figure 5—figure supplement 1C***). Together, these results indicate that MLL4 co-localizes with lineage-determining TFs on active enhancers during differentiation.

## MLL3 and MLL4 are major H3K4 mono- and di-methyltransferases in cells

The enrichment of H3K4 methyltransferase MLL4 on active enhancers, which are marked by H3K4me1/2 and H3K27ac, prompted us to examine the enzymatic properties of MLL4. For this purpose, we affinity-purified endogenous SET1A/SET1B complex (SET1A/B.com) and MLL3/MLL4 complex

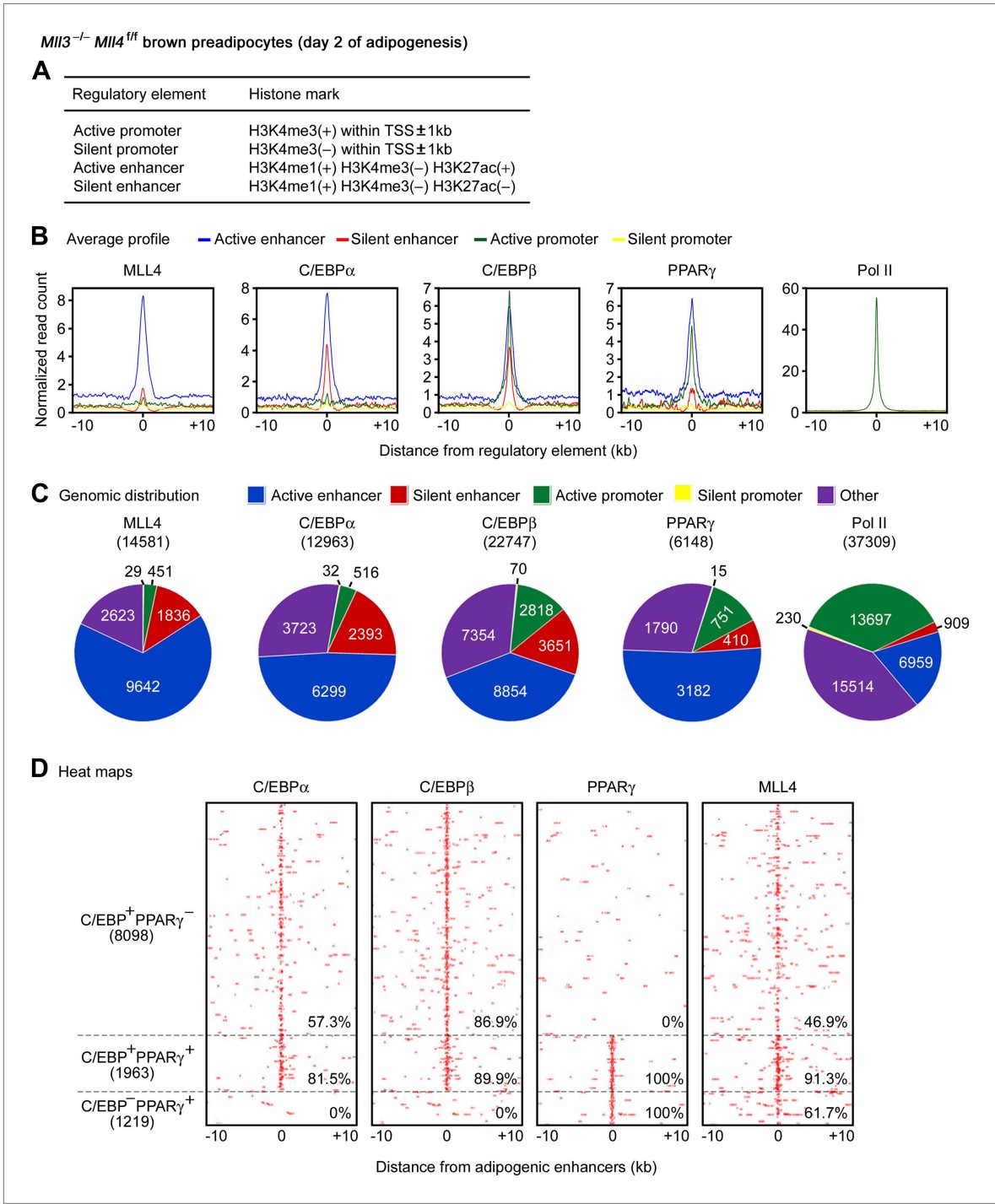

**Figure 5**. MLL4 co-localizes with lineage-determining TFs on active enhancers during differentiation. ChIP-Seq analyses of MLL4, TFs, H3K4me1/2/3 and H3K27ac were done at day 2 of adipogenesis. (**A**) Table depicting histone modifications used to define gene regulatory elements. TSS, transcription start site. (**B** and **C**) MLL4 is mainly localized on active enhancers during adipogenesis. (**B**) Average binding profiles of MLL4, adipogenic TFs C/EBPα, C/EBPβ and PPARγ, and RNA polymerase II (Pol II) around the center of each type of gene regulatory elements. (**C**) Pie charts depicting the genomic distributions of MLL4, C/EBPα, C/EBPβ, PPARγ and Pol II binding regions. (**D**) MLL4 co-localizes with adipogenic TFs on active enhancers during adipogenesis. The binding profiles of C/EBPα, C/EBPβ, PPARγ and MLL4 on the three types of adipogenic enhancers (C/EBP⁺PPARγ⁻, C/EBP⁻PPARγ⁺ and C/EBP⁺PPARγ⁺) are shown in heat maps. Adipogenic enhancers are defined as active enhancers bound with C/EBPα, C/EBPβ or PPARγ at day 2 of adipogenesis.

The following figure supplements are available for figure 5:

**Figure supplement 1**. MLL4 co-localizes with lineage-determining TFs on active enhancers during myogenesis.

(MLL3/4.com) from 293 cells expressing FLAG-tagged CFP1 and PA1, the unique subunit of SET1A/B. com and MLL3/4.com, respectively (*Lee and Skalnik, 2005*; *Cho et al., 2007*). In an in vitro HMT assay using core histones as substrate, we found that compared with SET1A/B.com, the major H3K4 tri-methyltransferase in mammalian cells, MLL3/4.com carried much stronger mono- and di-methyltransferase activities but much weaker tri-methyltransferase activity on H3K4 (*Figure 6A*). Time-course HMT assays confirmed H3K4me1/2 methyltransferase activity of MLL3/4.com in vitro (*Figure 6B*).

In *Mll3* single KO brown preadipocytes, we observed a moderate decrease of H3K4me1 (*Figure 6—figure supplement 1A*). Deletion of *Mll4* from *Mll3⁻/⁻Mll4^{f/f}* cells led to global decreases of H3K4me1 and H3K4me2 but much less effect on H3K4me3 levels at day 0 and day 2 of adipogenesis. H3K27ac levels also decreased significantly (*Figure 6C*). In *Mll3/Mll4* double KO myocytes and human colon cancer cells (*Guo et al., 2012*), we also observed significant decreases of H3K4me1 (*Figure 6—figure supplement 1B* and *Figure 6D*). ChIP-Seq analyses of MLL4 and histone modifications also revealed that among the 14,581 MLL4 binding sites during adipogenesis, 12,783 (87.7%) were marked by both H3K4me1 and H3K4me2 (*Figure 6E*). These data indicate that endogenous MLL3 and MLL4 are major H3K4 mono- and di-methyltransferases in mammalian cells.

## MLL4 is required for enhancer activation and cell-type-specific gene expression during differentiation

The enrichment of H3K4 mono- and di-methyltransferase MLL4 on active enhancers during differentiation prompted us to investigate whether MLL4 is required for H3K4me1/2 on these enhancers. ChIP-Seq

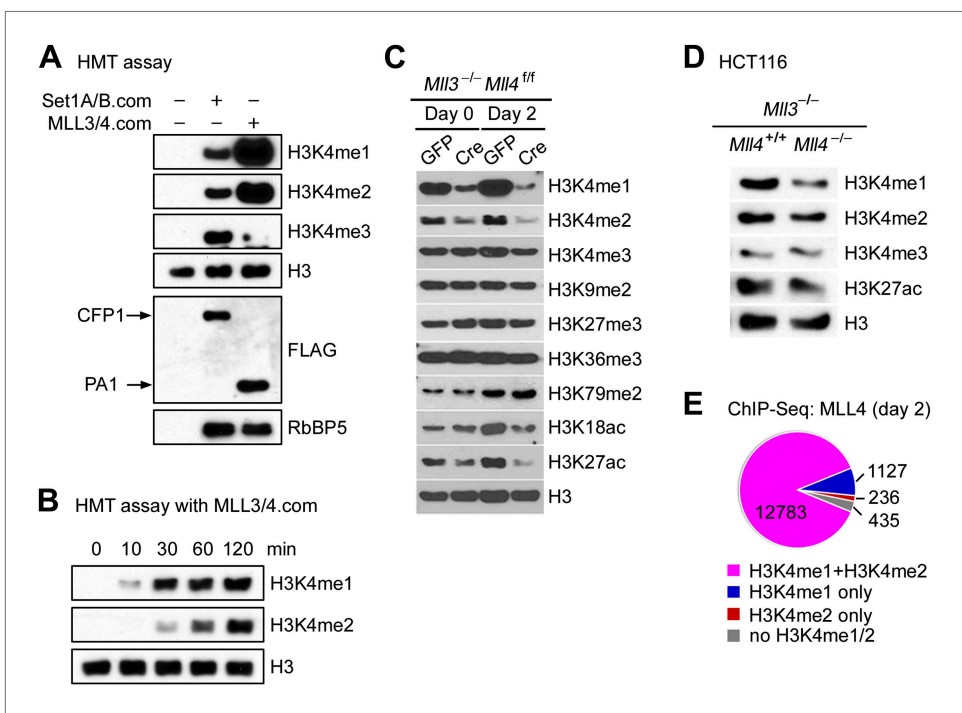

**Figure 6**. MLL4 is a major H3K4 mono- and di-methyltransferase in cells. (**A** and **B**) In vitro histone methyltransferase (HMT) assay. (**A**) SET1A/SET1B complex (SET1A/B.com) and MLL3/MLL4 complex (MLL3/4.com) that were affinity-purified from 293T nuclear extracts were incubated with core histones in an HMT assay for 3 hr, followed by Western blot analysis. (**B**) Time-course HMT assay with MLL3/4.com. (**C**) *Mll3⁻/⁻Mll4^{f/f}* brown preadipocytes were infected with adenoviral GFP or Cre as in *Figure 2*. Cells were collected at day 0 and day 2 of adipogenesis for Western blot analysis of histone modifications. (**D**) Western blot analysis of histone modifications in *MLL3⁻/⁻MLL4⁻/⁻* HCT116 human colon cancer cells. (**E**) 87.7% of MLL4-binding regions at day 2 of adipogenesis were marked by H3K4me1/2 as revealed by ChIP-Seq analyses of MLL4 and H3K4me1/2.

The following figure supplements are available for figure 6:

**Figure supplement 1**. MLL3 and MLL4 are H3K4 mono- and di-methyltransferases in mammalian cells.

analyses revealed a dramatic increase of MLL4 levels on the 6,342 MLL4$^+$ adipogenic enhancers from day 0 to day 2 of adipogenesis (*Figure 7A*). Consistently, we observed marked increases of H3K4me1/2 on these enhancers. H3K27ac, Mediator (represented by the MED1 subunit) and Pol II levels also increased markedly on MLL4$^+$ adipogenic enhancers from day 0 to day 2. Deletion of *Mll4* from *Mll3*$^{-/-}$*Mll4*$^{f/f}$ cells prevented the marked increases of not only H3K4me1/2 but also H3K27ac, Mediator and Pol II on the 6,342 MLL4$^+$ adipogenic enhancers from day 0 to day 2 (*Figure 7A*). On *Pparg* and *Cebpa* gene loci, deletion of *Mll4* also prevented the increases of H3K4me1/2, H3K27ac, Mediator and Pol II on MLL4$^+$ adipogenic enhancers during adipogenesis (*Figure 4—figure supplement 1 and 2*). On the 480 MLL4$^+$ promoters identified during adipogenesis, MLL4 was required for H3K4me1/2 but not H3K4me3 levels. However, the effect of *Mll4* deletion on promoter H3K4me1/2 is not as severe as on enhancers (*Figure 7—figure supplement 1* vs *Figure 7A*).

During myogenesis, MLL4 and MyoD levels increased dramatically on the 5,228 MLL4$^+$ myogenic enhancers. Deletion of *Mll4* from *Mll3*$^{-/-}$*Mll4*$^{f/f}$ cells also prevented the marked increases of H3K4me1 and H3K27ac on MLL4$^+$ myogenic enhancers during myogenesis (*Figure 7—figure supplement 2A-B*). Thus, MLL4 is the major H3K4 mono- and di-methyltransferase on adipogenic and myogenic enhancers. Because H3K27ac is a mark for active enhancers, these results indicate that MLL4 is required for the activation of cell-type-specific enhancers during adipogenesis and myogenesis.

We next asked how MLL4 affects the induction and expression of genes regulated by cell-type-specific enhancers. To focus on the direct effect of MLL4, we examined genes associated with MLL4$^+$ adipogenic enhancers (C/EBP$^+$PPARγ$^-$, C/EBP$^-$PPARγ$^+$, or C/EBP$^+$PPARγ$^+$). We first looked at the induction of genes from day 0 to day 2 of adipogenesis (*Figure 7B*). Genes associated with MLL4$^+$ adipogenic enhancers at day 2 were better induced than those associated with MLL4$^-$ adipogenic enhancers (p=1.5E-22, Wilcoxon test). The induction was even stronger if the associated MLL4$^+$ adipogenic enhancers were C/EBP$^+$PPARγ$^+$ (p=1.5E−76, Wilcoxon test). We then examined how MLL4 affects expression of genes at day 2 of adipogenesis. As shown in *Figure 7C*, deletion of *Mll4* significantly decreased expression of genes associated with MLL4$^+$ adipogenic enhancers (C/EBP$^+$PPARγ$^-$ or C/EBP$^-$PPARγ$^+$, p=2.2E−10, Wilcoxon test). An even stronger effect of *Mll4* deletion was observed if the associated MLL4$^+$ adipogenic enhancers were C/EBP$^+$PPARγ$^+$ (p=1.8E−31, Wilcoxon test). In contrast, the deletion of *Mll4* had little effect on the expression of genes associated with MLL4$^-$ adipogenic enhancers. In myocytes, deletion of *Mll4* significantly decreased expression of genes associated with MLL4$^+$ myogenic enhancers but not those associated with MLL4$^-$ myogenic enhancers (*Figure 7—figure supplement 2C*). Together, these results indicate that MLL4 is required for activation of enhancers that are important for cell-type-specific gene expression during differentiation.

## C/EBPβ recruits and requires MLL4 to establish a subset of adipogenic enhancers

The physical interaction and the genome-wide co-localization of C/EBPβ with MLL4 during adipogenesis suggest that the early adipogenic TF C/EBPβ may recruit MLL4 to establish at least a subset of adipogenic enhancers. To directly test this possibility, we ectopically expressed C/EBPβ in preadipocytes (*Figure 8A*), followed by ChIP-Seq of C/EBPβ, MLL4, H3K4me1 and H3K27ac without inducing differentiation. Of the 4,965 C/EBPβ$^+$ MLL4$^+$ active enhancers identified at day 2 of adipogenesis, 66.6% (3,309/4,965) were bound by ectopic C/EBPβ in undifferentiated preadipocytes. Ectopic C/EBPβ was able to recruit MLL4 to a subset of enhancers (666 out of 3,309) (*Figure 8B*). Among them, 88.7% (591/666) showed the characteristics of active enhancers, as indicated by the presence of H3K4me1 and H3K27ac (*Figure 8C*). Among the 591 recovered C/EBPβ$^+$ MLL4$^+$ active enhancers, 375 displayed ectopic C/EBPβ-induced de novo MLL4 binding, which led to significant increases of H3K4me1 and H3K27ac. The remaining 216 recovered C/EBPβ$^+$ MLL4$^+$ adipogenic enhancers, which were pre-marked by MLL4 in the control cells, showed ectopic C/EBPβ-enhanced MLL4 binding as well as H3K4me1 and H3K27ac (*Figure 8C*). Importantly, deletion of *Mll4* markedly decreased H3K4me1 and H3K27ac on over 90% of the 591 recovered C/EBPβ$^+$ MLL4$^+$ adipogenic enhancers (*Figure 8C–D*). MLL4-dependent de novo H3K4me1 and H3K27ac were also observed on the C/EBPβ$^+$ adipogenic enhancers located on *Pparg* gene locus (*Figure 8E*). Together, these results suggest that lineage-determining TF C/EBPβ recruits and requires MLL4 to establish at least a subset of adipogenic enhancers.

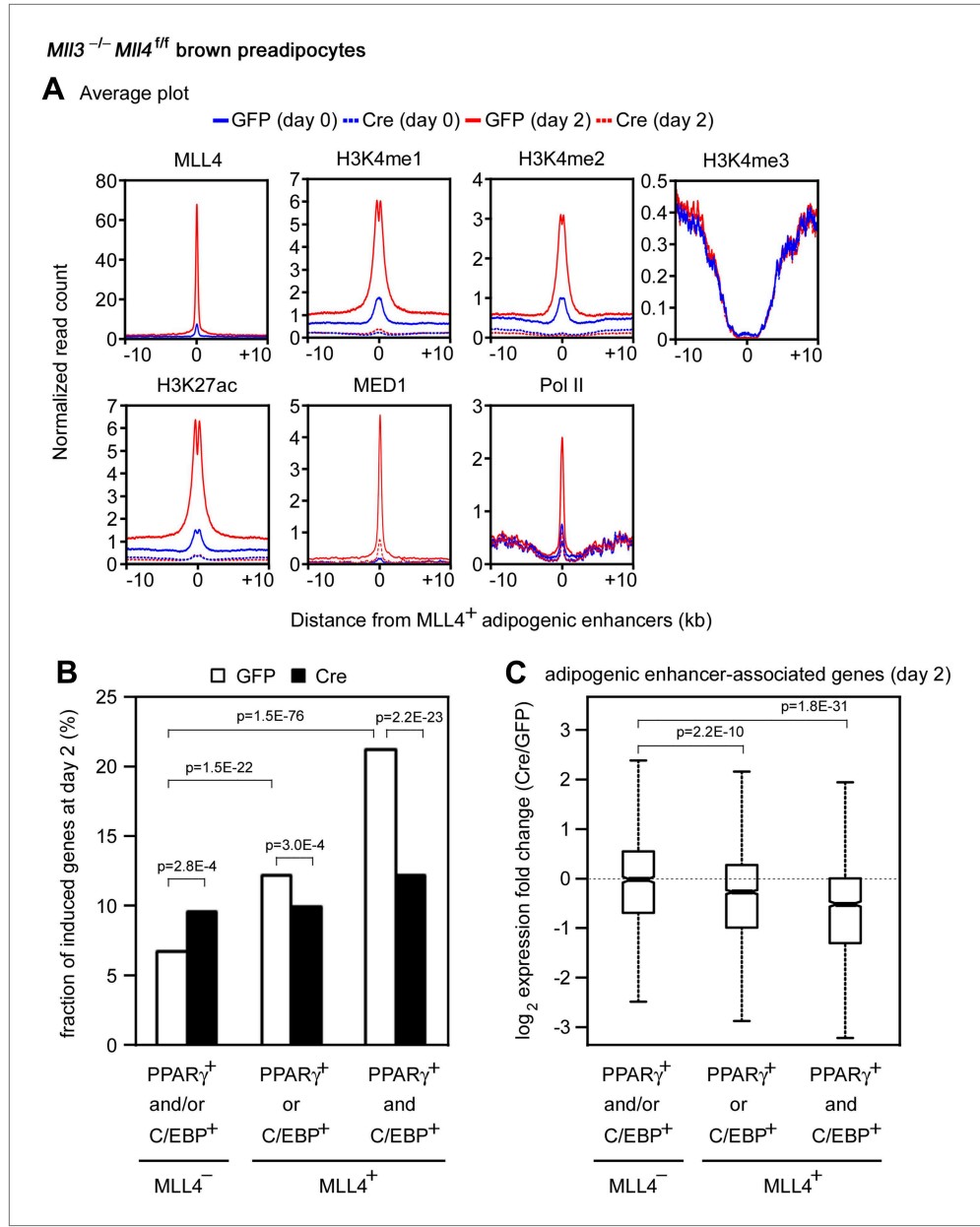

**Figure 7**. MLL4 is required for enhancer activation during differentiation. *Mll3⁻/⁻Mll4^f/f* brown preadipocytes were infected with adenoviral GFP or Cre as in *Figure 2*. Cells were collected at day 0 and day 2 of adipogenesis for ChIP-Seq of H3K4me1/2/3, H3K27ac, MED1 and Pol II, and RNA-Seq. (**A**) Deletion of *Mll4* dramatically decreases H3K4me1/2, H3K27ac, MED1 and Pol II levels on MLL4 positive (MLL4⁺) adipogenic enhancers. Average profiles of histone modifications, MED1 and Pol II on MLL4⁺ adipogenic enhancers are shown. (**B**) MLL4 promotes induction of genes associated with MLL4⁺ adipogenic enhancer during adipogenesis. (**C**) Deletion of *Mll4* reduces expression of MLL4⁺ adipogenic enhancer-associated genes. Gene expression fold changes were obtained by comparing Cre-infected with GFP-infected cells at day 2 of adipogenesis and are shown in the box plot.

The following figure supplements are available for figure 7:

**Figure supplement 1**. MLL4 is required for H3K4me1/2 on MLL4⁺ promoters.

**Figure supplement 2**. MLL4 is required for enhancer activation during myogenesis.

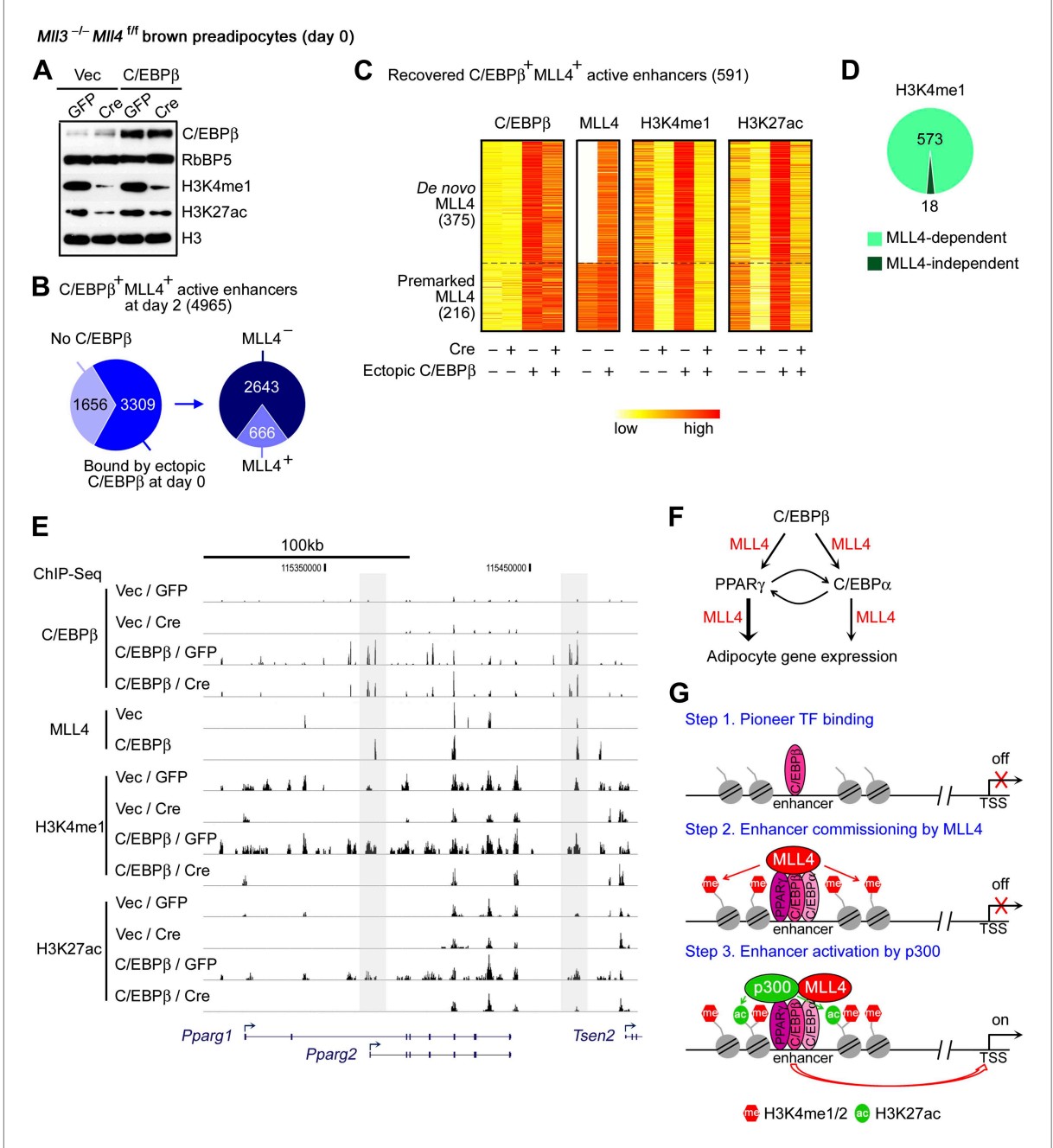

**Figure 8**. C/EBPβ recruits and requires MLL4 to establish a subset of adipogenic enhancers. *Mll3⁻/⁻ Mll4^f/f* brown preadipocytes were infected with retroviral C/EBPβ or Vec only, and then infected with adenoviral GFP or Cre. Cells were collected 2 days after confluence without induction of differentiation for Western blot (**A**) and for ChIP-Seq of C/EBPβ, MLL4, H3K4me1 and H3K27ac (**B–E**). (**A**) Western blot analyses of C/EBPβ expression and histone modifications. Nuclear protein RbBP5 was used as a loading control. (**B**) Ectopic expression of C/EBPβ in undifferentiated preadipocytes leads to MLL4 binding on a subset of C/EBPβ⁺ MLL4⁺ active enhancers identified at day 2 of adipogenesis. (**C**) Heat maps of the relative signals of C/EBPβ, MLL4, H3K4me1 and H3K27ac on the recovered C/EBPβ⁺ MLL4⁺ active enhancers. (**D**) MLL4 is required for H3K4me1 on the recovered C/EBPβ⁺ MLL4⁺ active enhancers. (**E**) Genome browser view of ectopic C/EBPβ-induced MLL4 binding as well as H3K4me1 and H3K27ac on *Pparg* locus. (**F**) Model depicting the role of MLL4 in transcriptional regulation of adipogenesis. The early adipogenic TF C/EBPβ serves as a pioneer TF and recruits MLL4 to establish active enhancers on *Pparg* and *Cebpa* loci. After PPARγ and C/EBPα are induced, they recruit MLL4 and cooperate with other adipogenic TFs to shape the enhancer landscape important for adipocyte gene expression. (**G**) Model depicting the role of MLL4 in the step-wise enhancer activation.

The following figure supplements are available for figure 8:

**Figure supplement 1**. Time-course ChIP-qPCR on C/EBPβ⁺MLL4⁺ enhancers on *Pparg* gene locus.

Consistently, by quantitative time-course ChIP assays performed during the first 48 hr of adipogenesis, we observed sequential enrichments of lineage-determining TF C/EBPβ, H3K4me1/2 methyltransferase MLL4, H3K4me1/2, and H3K27ac on C/EBPβ⁺MLL4⁺ adipogenic enhancers identified on *Pparg* gene locus (*Figure 8—figure supplement 1*). Combined with our earlier observations, these results suggest a step-wise model of enhancer activation during cell differentiation (*Figure 8G* and 'Discussion').

## Discussion

In this paper, we identify MLL4 as a major H3K4 mono- and di-methyltransferase enriched on mammalian enhancers. We show MLL4 is essential for enhancer activation, cell-type-specific gene expression, and cell differentiation. Our findings also shed new light on how histone modifying enzymes coordinate with lineage-determining transcription factors on enhancers to regulate gene expression and cell differentiation.

### MLL4 is a major H3K4me1/2 methyltransferase on mammalian enhancers

*Drosophila* Trr and its mammalian homologs MLL3/MLL4 represent the major candidate methyltransferases for H3K4me1 on enhancers (*Calo and Wysocka, 2013*). Our in vitro HMT assays show that MLL3/4.com carries predominantly H3K4 mono- and di-methyltransferase activity with little H3K4 tri-methyltransferase activity, in agreement with two recent studies (*Tang et al., 2013*; *Wu et al., 2013*). By generating *Mll3/Mll4* double KO cells, we demonstrate that endogenous MLL3 and MLL4 are major H3K4me1/2 methyltransferases in mammalian cells and that MLL4 has a partial functional redundancy with MLL3 in cells.

False-positive signals due to potential cross-reactivity of antibodies are a major concern in ChIP-Seq analyses (*Kidder et al., 2011*). To exclude false-positive signals and identify bona fide MLL4 binding regions, we performed ChIP-Seq of MLL4 in both *Mll3* KO and *Mll3/Mll4* double KO cells. The results provide the direct evidence that MLL4 is a major H3K4me1/2 methyltransferase on enhancers during cell differentiation. More importantly, our data indicate that MLL4 is required for enhancer activation, which suggests that H3K4me1/2 may be important for enhancer activation. The partial functional redundancy between MLL3 and MLL4 in cells suggests that endogenous MLL3 is likely an H3K4me1/2 methyltransferase on enhancers and is likely involved in the activation of cell-type-specific enhancers.

Significant levels of H3K4me1/2 remain in *Mll3/Mll4* double KO cells (*Figure 6C*), indicating that other H3K4 methyltransferases contribute to H3K4me1/2 in cells. Studies have shown possible roles of MLL1 and Set7 in catalyzing H3K4me1 on specific enhancers (*Blum et al., 2012*; *Kaikkonen et al., 2013*), but whether these enzymes are required for enhancer activation remains to be determined. It was shown recently that knockdown of several H3K4 methyltransferases including MLL1, MLL3, MLL4 in macrophages led to significant decreases of H3K4me1/2 on latent enhancers (*Kaikkonen et al., 2013*). Our data indicate that MLL4 is required for the activation of cell-type-specific enhancers during differentiation. Whether MLL4 is also required for the activation of latent enhancers remains to be determined.

### Lineage-determining TFs recruit MLL4 to establish cell type-specific enhancers

During adipogenesis and myogenesis, MLL4 exhibits cell-type- and differentiation-stage-specific genomic binding and co-localizes with lineage-determining TFs and H3K4me1/2 on active enhancers. Thus, MLL4 appears to mark lineage-determining enhancers during differentiation. It will be interesting to test whether MLL4 binding can predict enhancers in other cell types. Interestingly, MLL4-dependent H3K4me1/2 distribute well beyond the binding sites of lineage-determining TFs and MLL4. A similar observation can be made for p300 and H3K27ac (*Rada-Iglesias et al., 2011*), where H3K27ac is distributed much broader than p300. We speculate that lineage-determining TFs along with MLL4 may induce physical proximity of neighboring nucleosomes to MLL4, which enables MLL4-mediated H3K4me1/2 over a much broader region.

C/EBPβ is a lineage-determining TF for adipogenesis. It can bind to relatively closed chromatin and thus behaves as a pioneer factor in adipogenesis (*Siersbaek et al., 2011*). By ectopic expression of C/EBPβ

in preadipocytes without inducing differentiation, we show that C/EBPβ recruits and requires MLL4 to establish a subset of adipogenic enhancers. Our data are consistent with earlier reports that genomic binding of the lineage-determining TF PU.1 leads to H3K4me1 in macrophages and B cells and that ectopic expression of PU.1 in non-hematopoietic fibroblasts induces H3K4me1 on macrophage-specific enhancers (*Ghisletti et al., 2010*; *Heinz et al., 2010*). Although it remains to be determined whether MLL4 and/or MLL3 play a major role in establishing enhancers in macrophages, B cells and other cell types, the available results suggest that lineage-determining TFs recruit H3K4me1/2 methyltransferases to prime enhancer-like regions in a particular cell type.

Ectopic expression of C/EBPβ in preadipocytes without differentiation only recovers a subset of endogenous C/EBPβ⁺MLL4⁺ enhancers, indicating that the recruitment of MLL4 to enhancers during adipogenesis involves additional mechanisms and factors. Because differentiation-dependent phosphorylation of C/EBPβ increases its DNA binding activity, the optimal genomic binding of C/EBPβ and the subsequent recruitment of MLL4 are likely dependent on induction of adipogenesis. Similarly, the binding of MyoD to myogenic enhancers on *Myog* locus is differentiation-dependent (*Figure 7—figure supplement 2B*). The direct interaction between MLL3/4.com and PPARγ and the genomic co-localization of MLL4 with PPARγ and C/EBPα suggest that PPARγ and C/EBPα also play critical roles in recruiting MLL4 to establish adipogenic enhancers.

Motif analyses of MLL4 binding sites during adipogenesis and myogenesis have identified motifs of additional lineage-determining TFs, such as adipogenic TFs EBFs, GR and NFIC and myogenic TFs TCF3, Runx1 and TEAD4 (*Figure 4A*). Future ChIP-Seq analyses will tell whether these additional lineage-determining TFs can recruit MLL4 to regions that are not bound by C/EBPα/β, PPARγ or MyoD. On the other hand, MLL4 is absent from many genomic regions that are occupied by C/EBPα/β or PPARγ during adipogenesis or by MyoD in myocytes (*Figure 4B,D*), suggesting that lineage-determining TFs alone are not always sufficient for recruiting MLL4 to establish enhancers and that additional mechanisms and factors may be involved.

Ectopic expression of C/EBPβ in preadipocytes partially mimics the early phase of adipogenesis when PPARγ and C/EBPα have not been induced yet. Nevertheless, endogenous C/EBPβ co-localizes with MLL4 on active enhancers located on both *Pparg* and *Cebpa* gene loci (*Figure 4—figure supplements 1 and 2*). Because C/EBPβ not only transcriptionally activates *Pparg* and *Cebpa* gene expression but also facilitates PPARγ and C/EBPα genomic binding during adipogenesis (*Rosen et al., 2002*; *Siersbaek et al., 2011*), our data suggest the following model on how MLL4 regulates adipogenesis (*Figure 8F*). The pioneer TF C/EBPβ recruits MLL4 to activate adipogenic enhancers on, and the subsequent induction of, *Pparg* and *Cebpa* gene expression. After the master adipogenic TFs PPARγ and C/EBPα are induced, they cooperate to recruit MLL4 to establish the downstream enhancer landscape critical for adipocyte gene expression.

In cell culture, MLL3 and MLL4 are partially redundant in adipogenesis. In mice, deletion of *Mll4* alone causes severe defects in BAT development. Deletion of *Mll3* in mice has been shown to decrease white adipose tissue size (*Lee et al., 2008*). These results suggest that while MLL4 is the dominant one during mouse embryonic development, MLL3 is partially redundant with MLL4 in mice. The critical roles of MLL3 and MLL4 in adipogenesis are consistent with our previous report that PTIP, a component of the MLL3/4.com, is essential for *Pparg* and *Cebpa* expression and adipogenesis (*Cho et al., 2009*).

## How does MLL4 work on enhancers?

The presence of H3K4me1 on enhancers often precedes the active enhancer mark H3K27ac and gene expression, suggesting that H3K4me1 broadly defines a window of future active enhancers (*Calo and Wysocka, 2013*). Consistently, our time-course ChIP assays revealed sequential appearances of H3K4me1/2 and H3K27ac on C/EBPβ⁺MLL4⁺ active enhancers on *Pparg* gene locus in the early phase of adipogenesis (*Figure 8—figure supplement 1*). Further, we show that the H3K4me1/2 methyltransferase MLL4 is required for H3K27ac, Mediator and Pol II on enhancers, which indicates that MLL4 is required for enhancer activation and suggests that MLL4 exerts its function in enhancer commissioning through H3K4me1/2. Collectively, our data suggest a stepwise model of enhancer activation during adipogenesis (*Figure 8G*). Step 1, pioneer TF binding. Pioneer TFs such as C/EBPβ bind enhancer-like regions. Step 2, enhancer commissioning by MLL4. Lineage-determining TFs, such as C/EBPβ, PPARγ and C/EBPα, cooperatively recruit MLL4 to perform H3K4me1/2 on enhancer-like regions. Step 3, enhancer activation by H3K27 acetyltransferase p300,

followed by Pol II recruitment, establishment of enhancer–promoter interaction, and activation of cell-type-specific gene expression. Thus, the enhancer-associated H3K4me1/2 methyltransferase MLL4 appears to perform the opposite function of LSD1, an H3K4me1/2 demethylase required for decommissioning of embryonic stem (ES) cell-specific enhancers during ES cell differentiation (*Whyte et al., 2012*).

### MLL3/MLL4 mutations in diseases

Frequent loss-of-function mutations in MLL4 (sometimes called MLL2) and its homolog MLL3 have been identified in developmental diseases such as Kabuki syndrome (*Ng et al., 2010*), congenital heart disease (*Zaidi et al., 2013*), and in cancers such as medulloblastoma (*Parsons et al., 2011*; *Jones et al., 2012*; *Pugh et al., 2012*), non-Hodgkin lymphomas (*Morin et al., 2011*; *Pasqualucci et al., 2011*), breast cancer (*Ellis et al., 2012*), and prostate cancer (*Grasso et al., 2012*). Our findings suggest that loss-of-function mutations in MLL3 and MLL4 would impair H3K4me1/2 on enhancers, which lead to defects in enhancer activation, cell-type-specific gene expression and cell differentiation. Such a mechanism may contribute to the pathogenesis of these developmental diseases and cancers.

## Materials and methods

### Plasmids and antibodies

The retroviral plasmids pMSCVhygro-PPARγ2, pWZLhygro-C/EBPβ, and MSCVpuro-Cre have been described (*Ge et al., 2008*; *Wang et al., 2010*). MyoD cDNA was subcloned from MSCVpuro-MyoD into pWZLhygro. The shRNA sequence-targeting mouse *Mll4* gene (GCATGTTCTTCAAGGACAAGA) was cloned into lentiviral vector pLKO.1.

The following homemade antibodies have been described: anti-UTX (*Hong et al., 2007*); anti-PTIP, anti-PA1#2, anti-MLL4#3 (*Cho et al., 2009*). Anti-C/EBPα (sc-61X), anti-C/EBPβ (sc-150X), anti-PPARγ (sc-7196X), and anti-MyoD (sc-760) were from Santa Cruz Biotechnology (Dallas, TX, USA). Anti-Menin (A300-105A), anti-RbBP5 (AA300-109A), and anti-MED1/TRAP220 (A300-793A) were from Bethyl Laboratories (Montgomery, TX, USA). Anti-H3K4me1 (ab8895) was from Abcam (Cambridge, MA, USA). Anti-Pol II (17-672) and anti-H3K4me3 (07-473) were from Millipore (Billerica, MA, USA). Other histone methylation and acetylation antibodies have been described (*Jin et al., 2011*).

### Generation of mouse strains

*Mll3*[+/−] and *Mll4*[+/−] mice were generated from BayGenomics gene trap ES cell lines following standard procedures at NIDDK Mouse Knockout Core (*Stryke et al., 2003*). *Mll3*[+/−] mice were derived from ES cell line XM083, in which the gene trap vector pGT0lxf was inserted between exon 9 and 10 of one *Mll3* allele. *Mll4*[+/−] mice were derived from ES cell line XT0709, in which the gene trap vector pGT0lxf was inserted between exon 19 and 20 of one *Mll4* allele.

To generate *Mll4* conditional KO mice, the loxP/FRT-flanked neomycin cassette was inserted at the 3′ of exon 19 and the single loxP site was inserted at the 5′ of exon 16. The targeted region includes exons 16–19 (*Figure 1A*). The *Mll4*[floxneo/+] ES cell was injected into blastocysts to obtain male chimera mice that were crossed with wild-type C57BL/6J females to screen for germ line transmission. Mice bearing germ-line transmission (*Mll4*[floxneo/+]) were crossed with FLP1 mice (Jackson no. 003946) to generate *Mll4*[f/+] (i.e., *Mll4*[flox/+]) mice. *Mll4*[f/+] mice were crossed with *Myf5-Cre* (Jackson no. 007893). The resulting *Mll4*[f/+];*Myf5-Cre* were then crossed with *Mll4*[f/f] to generate *Mll4*[f/f];*Myf5-Cre* mice. Deletion of exons 16–19 by Cre causes open reading frame shift and creates a stop codon in exon 20. The resulting *Mll4* KO allele encodes a truncated MLL4 protein lacking the C-terminal ~4200aa. For genotyping the *Mll4* alleles, allelic PCR was developed as shown in *Figure 1A–B*. P1:5′-GTTCACTCAGTGGGGCTGTG-3′; P2: 5′-ATTGCATCAGGCAAATCAGC-3′; P3: 5′-GCAGAAGCCTGCTATGTCCA-3′. Genotyping of Myf5-Cre was done by PCR using the following three primers: 5′-CGTAGACGCCTGAAGAAGGTCAACCA-3′, 5′-CACATTAGAAAACCTGCCAACACC-3′, and 5′-ACGAAGTTATTAGGTCCCTCGAC-3′. PCR amplified wild-type (603 bp) and *Myf5-Cre* allele (400 bp).

All mouse work was approved by the Animal Care and Use Committee of NIDDK, NIH.

### Histology and immunohistochemistry

E18.5 embryos were dissected out by Cesarean section and fixed in 4% paraformaldehyde overnight at 4°C. The embryos were further dehydrated and embedded in paraffin and sectioned at 7–10 μm

with a microtome. H&E staining and immunohistochemistry (IHC) on paraffin sections were done as described (*Feng et al., 2010*). The primary antibodies used for IHC were 1:20 dilution of anti-Myosin (MF20; Developmental Studies Hybridoma Bank) and 1:400 dilution of anti-UCP1 (ab10983; Abcam). Fluorescent secondary antibodies used were Alexa Fluor 488 goat anti-mouse IgG2b and Alexa Fluor 555 goat anti-rabbit IgG (Life Technologies, Carlsbad, CA, USA).

### Immortalization of primary brown preadipocytes, virus infection, adipogenesis and myogenesis assays

Primary brown preadipocytes were isolated from interscapular BAT of E18.5 embryos and were immortalized with SV40T-expressing retroviruses pBabepuro-large T as described (*Wang et al., 2010*). Adenoviral infection of preadipocytes was done at 50 moi. Adipogenesis of immortalized brown preadipocytes and 3T3-L1 white preadipocyte cell line was done as described (*Wang et al., 2013*). PPARγ- or C/EBPβ-stimulated adipogenesis and MyoD-stimulated myogenesis were done as described (*Ge et al., 2002*). Myogenesis was induced in near-confluent cells.

### qPCR

Total RNA extraction and qRT-PCR were done as described (*Cho et al., 2009*). Taqman probe for *Mll3* (assay ID Mm01156965_m1) was from Life Technologies. qRT-PCR of *Mll4* was done using SYBR green primers: forward 5'-GCTATCACCCGTACTGTGTCAACA-3' and reverse 5'-CACACACGATACAC TCCACACAA-3'. Taqman probes for *Pparg1* and *Pparg2* and other SYBR green primers for qRT-PCR have been described (*Wang et al., 2013*). SYBR green primers for ChIP-qPCR are listed in *Supplementary file 1B*.

### ChIP-Seq and RNA-Seq

ChIP was performed by following a protocol from Myers' laboratory (http://www.hudsonalpha.org/myers-lab/protocols) with modifications. The cells were crosslinked with 1–2% formaldehyde for 10 min at room temperature. Crosslinking reaction was stopped by adding 125 mM glycine. The cells were washed with cold PBS twice. $2 \times 10^7$ cells were collected in 10 ml Farnham lysis buffer (5 mM PIPES pH 8.0/85 mM KCl/0.5% NP-40, supplemented with protease inhibitors) and centrifuged at 4,000 *g* for 5 min at 4°C. Cell pellet was washed with 10 ml Farnham lysis buffer, followed by centrifugation. Resulting nuclear pellet was resuspended in 1 ml TE buffer (10 mM Tris-Cl pH 7.7/1 mM EDTA, supplemented with protease inhibitors) and sonicated for 17 min (30 s on/off cycle). Lysates were supplemented with detergents to make 1X RIPA buffer (10 mM Tris-Cl pH 7.7/1 mM EDTA/0.1% SDS/0.1% Na-DOC/1% triton X-100) and centrifuged to remove debris. For each ChIP, 6–8 µg antibodies were prebound to 50 µl Dynabeads Protein A (100.02D; Life Technologies) overnight at 4°C. Next day, antibody-beads complex was added to chromatin from $2 \times 10^7$ cells and further incubated overnight at 4°C. The beads were washed twice with RIPA buffer, twice with RIPA + 0.3M NaCl, twice with LiCl buffer (50 mM Tris-Cl pH 7.5/250 mM LiCl/0.5% NP-40/0.5% Na-DOC) and twice with PBS. DNA was eluted and reverse crosslinked in 200 µl elution buffer (1% SDS/0.1M NaHCO$_3$, supplemented with 20 µg proteinase K) overnight at 65°C. DNA was purified by QIAquick PCR Purification Kit (QIAGEN) and quantified. DNA and libraries were constructed as described (*Wei et al., 2012*). All ChIP-Seq and RNA-Seq samples were sequenced on Illumina HiSeq 2000.

### Computational analysis

#### Summary of computational analysis

Identification of ChIP-enriched regions was performed using the island approach 'SICER' (*Zang et al., 2009*). Gene ontology (GO) study was carried out using DAVID (*Huang et al., 2009*). Motif search around MLL4 binding regions was conducted using SeqPos (*He et al., 2010*).

#### Illumina pipeline analysis for ChIP-Seq and RNA-Seq data

The sequence reads were of 50bp in length and aligned to reference genome assembly NCBI37/mm9. The output of the Illumina Analysis Pipeline was converted to browser extensible data (BED) files detailing the genomic coordinates of each mapped read. To visualize the data on the UCSC genome browser (*Karolchik et al., 2008*), reads were collected and converted to wiggle (WIG) format using in-house script. The numbers of mapped reads are listed in *Supplementary file 1A*.

## Gene expression calculation using RNA-Seq data

For RNA-Seq datasets, we collected only reads that locate on exons as annotated in NCBI Reference Sequence Database (RefSeq) for assembly mm9, and calculated reads per kilobase per million (RPKM) for each gene as measure of expression level. Genes with RNA-Seq exonic RPKM>1 were regarded as expressed. In comparing gene expression levels before (D0, day 0) and during (D2, day 2) adipogenesis (or comparing gene expression levels in preadipocyte vs after myogenesis), we set a fold change cutoff of 2.5 to identify up-regulated and down-regulated genes. MLL4-dependent genes were defined as genes with over 2.5-fold down-regulation in MLL4-deficient cells ($Mll3^{-/-}Mll4^{-/-}$) compared with control cells ($Mll3^{-/-}$) (*Figure 2D,I*). Out of 14,902 genes expressed at D0 or D2 in adipogenesis, 1,531 were MLL4-dependent. Meanwhile, in the 1,302 up-regulated genes, 588 were MLL4-dependent. The p-value for the enrichment of MLL4-dependent genes in up-regulated genes was 1.4E−268 using hypergeometric test. The same approach was used for calculating the significance of MLL4-dependent induction in myogenesis. Genome-wide, 1,155 out of 16,805 expressed genes were MLL4-dependent. On the other hand, 836 out of 2,774 up-regulated genes were MLL4-dependent. The significance of MLL4-dependency in upregulated genes was $p<1E−300$ using hypergeometric test. GO studies were carried out using DAVID with the whole genome as background, and for each group of genes the GO terms with $p\leq10^{-7}$ were listed (*Figure 2E,J*).

## Identification of ChIP-enriched regions

To eliminate background noise in ChIP-Seq datasets, we determined ChIP-enriched regions for each ChIP-Seq sample using the SICER method (*Zang et al., 2009*). In particular, to eliminate non-specific binding of MLL4 antibody, we compared MLL4 ChIP-Seq sample from MLL4-deficient ($Mll3^{-/-}Mll4^{-/-}$) cells against that from control ($Mll3^{-/-}$) cells, and kept only the identified MLL4 binding regions with enrichment level significantly higher in control cells than in the MLL4-deficient cells, with an estimated false discovery rate (FDR) of less than $10^{-15}$. The same FDR threshold was also applied to C/EBPα and C/EBPβ ChIP-Seq samples when analyzed against the input libraries. For PPARγ ChIP-Seq at D0 (day 0) and D2 (day 2) of brown adipogenesis, MLL4 ChIP-Seq in C/EBPβ over-expressing brown preadipocytes, and MLL4 and MyoD ChIP-Seq in myocytes, the FDR threshold was set to be $10^{-5}$ due to the limited coverage of sequenced reads. For the ChIP-Seq datasets of histone modifications (H3K4me1/2/3 and H3K27ac), the window size was chosen to be 200 bp and the FDR threshold was chosen to be $10^{-3}$. For the ChIP-Seq datasets of MLL4, Pol II, MyoD and adipogenic factors (C/EBPα, C/EBPβ, and PPARγ), the window size was chosen to be 50 bp. Another required parameter in SICER, gap size, was chosen to be equal to 1 window.

## Genome-wide co-localization study

Specific MLL4-enriched regions identified at D0, D2, D7 (day 0, 2, 7) during adipogenesis as well as in myocytes were compared, and co-localization was defined as two regions overlapping for at least 1bp (*Figure 3A–B*). All D0 MLL4 regions, D2 MLL4 regions that did not overlap with D0 regions, D7 MLL4 regions that didn't overlap with either D0 or D2, and identified MLL4 peaks in myocytes but not in preadipocytes, were regarded as emergent MLL4 binding sites. Top 2,000 in each group as ranked by FDR against MLL4-deficient control were used later for GO and motif analyses. Each of the top 2,000 emergent MLL4 binding sites was assigned to its closest transcription start sites (TSS). The resulting gene list was imported into DAVID for GO analysis. The enriched GO terms in biological process with $p<10^{-5}$ were listed (*Figure 3D*). The same top emergent MLL4 sites were used as input to SeqPos (*He et al., 2010*) for motif search. All motifs found with calculated $p<10^{-10}$ were listed (*Figure 4A*). By the same 1bp-overlapping definition of co-localization, MLL4 binding sites at D2 were compared with C/EBPα, C/EBPβ, and PPARγ binding sites. Given the motif similarity and functional redundancy between C/EBPα and C/EBPβ, we first calculated the union of their binding sites and regarded them as C/EBPs (*Figure 4B*). The heat maps were generated with 50 bp resolution and ranked according to the intensity of MLL4 at the center of 400 bp window (*Figure 4C*). Using the same approach we checked co-localization between MLL4 and MyoD in myocytes (*Figure 4D–E*).

## Identification of enhancers and promoters during adipogenesis

By definitions described in *Figure 5A*, we identified enhancers and promoters along the genome. The definitions were applied in a mutually exclusive way, that is, for each genomic region, we first

checked whether it was promoter or enhancer (or others), then further pinned it down to active or silent ones (**Figure 5C**). The total length of the all active enhancers was 43% of total length of all enhancers and promoters. On the other hand, 9,642 of 11,948 (80.6%) MLL4 binding sites on enhancers and promoters were located on active enhancers. Therefore, the significance of MLL4 enrichment on active enhancers was p<1E−300 using binomial test. Average profiles were plotted using the number of ChIP-Seq reads (normalized to the size of each library) in 5 bp bins from the center of each regulatory element to 10 kb on both sides. The centers of regulatory elements were defined as the corresponding TSSs for promoters and the centers of H3K4me1 enriched regions for enhancers (**Figure 5B**). Furthermore, the active adipogenic enhancers were defined as C/EBPα, C/EBPβ, and PPARγ binding sites that were on active enhancers. Again C/EBP refers to the union of C/EBPα and C/EBPβ.

Genome-wide, out of the 26,348 C/EBP sites at D2, 8,235 (31.3%) were MLL4$^+$. On the other hand, out of the 10,061 C/EBP sites on active enhancers, 5,762 (57.3%) were MLL4$^+$. The significance for the preferential enrichment of the co-localization of MLL4 with C/EBP on active enhancers was p<1E−300 using hypergeometric test. Similarly, genome-wide, out of the 6,148 PPARγ sites at D2, 3,461 (56.3%) were MLL4$^+$. On the other hand, out of the 3,182 PPARγ sites on active enhancers, 2,544 (80.0%) were MLL4$^+$. The significance for the preferential enrichment of co-localization of MLL4 with PPARγ on active enhancers was p<1E−300 using hypergeometric test.

The heat maps were generated with 50 bp resolution and ranked according to the co-localization of C/EBPα, C/EBPβ, and PPARγ binding sites (**Figure 5D**). The 6,342 adipogenic enhancers that co-localize with MLL4 were used to profile various histone modifications, Pol II and MED1 in both control and MLL4-deficient cells (**Figure 7A**). Each active adipogenic enhancer was assigned to the nearest TSS within 1,000 kb, and the corresponding fraction of genes that were induced (fold change > 2.5) at D2 of adipogenesis and gene expression fold change (in base 2 logarithm) in KO cells were plotted using box plot, with outliers not shown to highlight the majority (**Figure 7B–C**). In **Figure 7B,C**, we classified genes associated with adipogenic enhancers into three mutually exclusive groups: C/EBP$^+$PPARγ$^-$, C/EBP$^-$PPARγ$^+$, or C/EBP$^+$PPARγ$^+$. A gene might be simultaneously associated with adipogenic enhancers with different patterns of TF occupation (i.e., C/EBP$^+$PPARγ$^-$, C/EBP$^-$PPARγ$^+$, or C/EBP$^+$PPARγ$^+$), resulting in ambiguity in designation. To resolve this ambiguity, we adopted the following rule: a promoter (and the corresponding gene) was categorized as C/EBP$^+$PPARγ$^+$ if at least one of its associated adipogenic enhancer was C/EBP$^+$PPARγ$^+$. A promoter was categorized as C/EBP$^+$PPARγ$^-$ or C/EBP$^-$PPARγ$^+$, if none of the associated adipogenic enhancer was C/EBP$^+$PPARγ$^+$, but at least one of them is C/EBP$^+$PPARγ$^-$ or C/EBP$^-$PPARγ$^+$. Similarly, a promoter is categorized as MLL4$^-$ if none of the associated adipogenic enhancer is MLL4$^+$.

## Identification of recovered C/EBPβ$^+$ active adipogenic enhancers in C/EBPβ over-expressing preadipocytes

Among the 4,965 C/EBPβ$^+$MLL4$^+$ active enhancers identified at D2 of adipogenesis, we found 3,309 of them had C/EBPβ binding in C/EBPβ over-expressing preadipocytes. 666 of 3,309 had MLL4 binding as well (**Figure 8B**). 591 of them also had H3K4me1 and H3K27ac enriched and were regarded as recovered C/EBPβ$^+$MLL4$^+$ active enhancers (**Figure 8C**). Based on the MLL4 enrichment in vector-expressing preadipocytes, we grouped recovered C/EBPβ$^+$MLL4$^+$ active enhancers into two categories, with MLL4 in vector-expressing preadipocytes (premarked) and without MLL4 in vector-expressing preadipocytes (de novo). We computed C/EBPβ, MLL4, H3K4me1 and H3K27ac enrichment intensities (as by normalized ChIP-Seq read count) in control and MLL4-deficient preadipocytes, with or without C/EBPβ over-expression (**Figure 8C**). We found that in both groups, H3K4me1 and H3K27ac were induced in C/EBPβ over-expressing cells, and the induction was MLL4-dependent: in using MLL4-deficient cells, over 95% of H3K4me1 islands corresponding to the recovered C/EBPβ$^+$ active adipogenic enhancers exhibited significant (FDR < 10$^{-3}$) decreased enrichment (**Figure 8D**).

## Identification of enhancers and promoters during myogenesis

By definitions described in **Figure 5A**, we identified enhancers and promoters along the genome. Then we checked the binding profile of MLL4 and MyoD on genomic regulatory elements to reveal the preferential binding pattern. Furthermore, we identified MyoD$^+$ active enhancers, and generated heat maps for MyoD and MLL4 at 50bp resolution (**Figure 5—figure supplement 1A-C**).

Genome-wide, 10,699 out of the 30,496 MyoD binding sites were MLL4⁺. On the other hand, 5,228 out of 8,091 MyoD binding sites on active enhancers were MLL4⁺. The significance for the preferential enrichment of co-localization of MLL4 with MyoD on active enhancers was p<1E−300 using hypergeometric test. The 5,228 active myogenic enhancers that co-localized with MLL4 were used to profile various histone modifications in both control and MLL4-deficient cells (*Figure 7—figure supplement 2A*). Each active myogenic enhancer was assigned to the nearest TSS within 1,000 kb, and the corresponding gene expression fold change (in base 2 logarithm) in MLL4-deficient cells were plotted using box plot, with outliers not shown to highlight the majority (*Figure 7—figure supplement 2C*).

## Accession numbers

All datasets described in this paper, including 8 RNA-Seq samples, 62 ChIP-Seq samples and 12 ChIP-Seq inputs, have been deposited in NCBI Gene Expression Omnibus under access # GSE50466.

## Acknowledgements

We thank Jeong-Heon Lee and David Skalnik for 293 cells stably expressing FLAG-CFP1, Cuiling Li and Chuxia Deng of NIDDK Mouse Knockout Core for generating chimera mice from ES cell lines, Keji Zhao, Harold Smith and NIDDK Genomics Core for sequencing, Yiping He for *MLL3/MLL4* double KO HCT116 cells. This work was supported by the Intramural Research Program of the NIDDK, NIH to KG

## Additional information

### Funding

| Funder | Grant reference number | Author |
|---|---|---|
| National Institute of Diabetes and Digestive and Kidney Diseases, National Institutes of Health | 1ZIADK075003 | Ji-Eun Lee, Chaochen Wang, Shiliyang Xu, Lifeng Wang, Anne Baldridge, Lenan Zhuang, Weiqun Peng, Kai Ge |
| National Institute of Arthritis, Musculoskeletal, and Skin Diseases, National Institutes of Health | 1ZIAAR041126 | Xuesong Feng, Vittorio Sartorelli |

The funders had no role in study design, data collection and interpretation, or the decision to submit the work for publication.

### Author contributions

J-EL, CW, WP, KG, Conception and design, Acquisition of data, Analysis and interpretation of data, Drafting or revising the article; SX, Acquisition of data, Analysis and interpretation of data, Drafting or revising the article; Y-WC, LW, XF, AB, VS, LZ, Acquisition of data, Analysis and interpretation of data

### Ethics

Animal experimentation: This study was performed in strict accordance with the recommendations in the Guide for the Care and Use of Laboratory Animals of the National Institutes of Health. All of the animals were handled according to approved institutional animal care and use committee (IACUC) protocols (K165-LERB-11) of NIDDK, NIH. All mouse work was approved by the Animal Care and Use Committee of NIDDK, NIH.

## Additional files

### Supplementary files

• Supplementary file 1. (**A**) Mapped read counts in RNA-Seq and ChIP-Seq. (**B**) List of SYBR Green primers for ChIP-qPCR

## Major dataset

The following dataset was generated:

| Author(s) | Year | Dataset title | Dataset ID and/or URL | Database, license, and accessibility information |
|---|---|---|---|---|
| Lee J-E, Wang C, Xu S, Peng W, Ge K | 2013 | H3K4 mono- and di-methyltransferase MLL4 is required for enhancer activation during cell differentiation | GSE50466; http://www.ncbi.nlm.nih.gov/geo/query/acc.cgi?token=qjszoecyprqfxut%26acc%3DGSE50466&acc=GSE50466 | Available at GEO database under the CC0 Public Domain Dedication (http://www.ncbi.nlm.nih.gov/geo/). |

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
