## [Decision Letter]

Thank you for sending your work entitled “H3K4 mono- and di-methyltransferase MLL4 is required for enhancer activation during cell differentiation” for consideration at *eLife*. Your article has been favorably evaluated by a Senior editor and 3 reviewers, one of whom, Christopher Glass, is a member of our Board of Reviewing Editors.

The Reviewing editor and the other reviewers discussed their comments before we reached this decision, and the Reviewing editor has assembled the following comments to help you prepare a revised submission.

The manuscript by Ge and colleagues reports on the molecular functions of MLL4 and MLL3 in mouse models of adipogenesis and myogenesis. Using conditional *MLL4* x systemic *MLL3* KO cells, they demonstrate that MLL4 is responsible for a significant fraction of the deposition of H3K4me1 and H3K4me2 at cell-specific enhancers required for adipogenesis and myogenesis. They further demonstrate that MLL4 localizes to these cell-specific enhancers, and that loss of MLL3/4 results in severe defects in cell-type specific gene expression and differentiation. They present evidence that MLL4 is recruited to enhancers required for adipogenesis by lineage determining factors C/EBP and PPARg. The authors conclude with a 3-step model of enhancer activation during adipogenesis.

All three reviewers found the work to be substantial and of high quality. There were some differences of opinion concerning novelty. One reviewer noted, “Many of these findings have already been investigated in previous studies. First, both MLL3 and MLL4 have been implicated in adipogenesis in a previous study by [30] (PNAS). Second, as acknowledged by the authors, it has already been established in several published studies such as the recent studies by [50] and [59] that the MLL family including MLL3 and MLL4 are preferentially H3K4me1 and H3K4me2 histone methyltransferases. In addition, this was very recently shown by Shilatifard and colleagues (Mol. Cell. Biol. doi:10.1128/MCB.01181-13.”) These points were discussed in the consultation session, with the conclusion being that the genome-wide findings related to the functional consequences of loss of MLL3/4 on enhancer activity and differentiation are novel and move the field forward significantly beyond the previous correlative findings.

Two major points would need to be addressed for the manuscript to be found acceptable for publication.

1) The three-step model of enhancer activation is not directly supported by the experiments because there is no evaluation of the intermediates in the model. All of the experiments evaluate two states: before and after differentiation. To establish evidence for the stepwise model, a time series is needed following initiation of the differentiation program that allows discrimination of the sequence of transcription factor binding, histone acetylation, H3K4me1/2, MLL4 binding, Pol II binding, etc. These studies could be performed using QT-PCR at informative loci.

2) More information is required to allow an assessment of data quality. The performance of ChIP-Seq for MLL4 in both WT and KO cells allays concerns as to possible false positives for this particular ChIP. However, the methods do not indicate whether biological replicates were performed for the sequencing based assays, raising questions regarding reproducibility. Since many of the analyses are based on overlaps defined by arbitrary thresholds, experimental variability could significantly affect quantitative aspects of the conclusions. If ChiP-Seq and RNA-Seq assays were only performed as single experiments, some form of independent validation of the key findings is required, , by QT-PCR of a representative set of informative genomic regions/RNAs. In addition, it would be helpful for the authors to provide a summary table for all of the sequencing experiments indicating the number of mapped reads that were used for analysis.

---

## [Author Response]

All three reviewers found the work to be substantial and of high quality. There were some differences of opinion concerning novelty. One reviewer noted, “Many of these findings have already been investigated in previous studies. First, both MLL3 and MLL4 have been implicated in adipogenesis in a previous study by [30] (PNAS). Second, as acknowledged by the authors, it has already been established in several published studies such as the recent studies by [50] and [59] that the MLL family including MLL3 and MLL4 are preferentially H3K4me1 and H3K4me2 histone methyltransferases. In addition, this was very recently shown by Shilatifard and colleagues (Mol. Cell. Biol. doi:10.1128/MCB.01181-13.”) These points were discussed in the consultation session, with the conclusion being that the genome-wide findings related to the functional consequences of loss of MLL3/4 on enhancer activity and differentiation are novel and move the field forward significantly beyond the previous correlative findings.

We greatly appreciate the favorable comments but we would like to clarify the findings reported in the literature.

First, while MLL3 has been implicated in adipogenesis, the underlying mechanism was largely unclear (30). MLL4 has not been implicated in adipogenesis and neither of MLL3/4 has been implicated in myogenesis. Using adipogenesis and myogenesis as model systems, we demonstrate that MLL3/4 function as H3K4me1/2 methyltransferases on enhancers to control cell-type-specific gene expression and cell differentiation.

Second, while our manuscript was in preparation, Tang et al. and Wu et al. reported that MLL3/4 are preferentially H3K4me1/2 methyltransferases using in vitro HMT assays. Our data confirm their in vitro findings. More importantly, we demonstrate using *MLL3/4* KO human and mouse cells that MLL3/4 are major mammalian H3K4me1/2 methyltransferases in vivo.

Third, the Shilatifard lab submitted their manuscript to *MCB* in early September. We submitted our manuscript to *eLife* also in early September. Our manuscript not only covers the main findings in the Shilatifard paper, but also shows that MLL4 is required for enhancer activation, cell-type-specific gene expression and differentiation.

*1) The three-step model of enhancer activation is not directly supported by the experiments because there is no evaluation of the intermediates in the model. All of the experiments evaluate two states: before and after differentiation. To establish evidence for the stepwise model, a time series is needed following initiation of the differentiation program that allows discrimination of the sequence of transcription factor binding, histone acetylation, H3K4me1/2, MLL4 binding, Pol II binding, etc. These studies could be performed using QT-PCR at informative loci*.

We have performed time-course ChIP-qPCR to examine the genomic binding of the pioneer TF C/EBPβ, MLL4, H3K4me1/2, H3K27ac and Pol II on three MLL4+ enhancer regions identified on *PPARγ* gene locus during the first 48h of adipogenesis of wild type brown preadipocytes (new Figure 8—figure supplement 1). The results indicate sequential bindings of C/EBPβ, MLL4, H3K4me1/2, H3K27ac and Pol II on these enhancers during adipogenesis, thus providing strong evidence to support the model in Figure 8.

Consistently with our data, Chris Glass and colleagues recently proposed a model in which H3K4me1/2 precedes H3K27ac during enhancer activation (Heinz et al., 2013).

*2) More information is required to allow an assessment of data quality. The performance of ChIP-Seq for MLL4 in both WT and KO cells allays concerns as to possible false positives for this particular ChIP. However, the methods do not indicate whether biological replicates were performed for the sequencing based assays, raising questions regarding reproducibility. Since many of the analyses are based on overlaps defined by arbitrary thresholds, experimental variability could significantly affect quantitative aspects of the conclusions. If ChiP-Seq and RNA-Seq assays were only performed as single experiments, some form of independent validation of the key findings is required, e.g., by QT-PCR of a representative set of informative genomic regions/RNAs. In addition, it would be helpful for the authors to provide a summary table for all of the sequencing experiments indicating the number of mapped reads that were used for analysis*.

We have confirmed ChIP-Seq data by ChIP-qPCR at MLL4+ enhancers on multiple adipogenesis genes (the new Figure 3—figure supplement 2). We have also confirmed RNA-Seq data by qRT-PCR (the new Figure 2—figure supplement 2).

We list in the new Supplementary File 1A the number of mapped reads in ChIP-Seq and RNA-Seq.